# Copy number signatures predict chromothripsis and clinical outcomes in newly diagnosed multiple myeloma

Kylee H. Maclachlan [1], Even H. Rustad[1,2], Andriy Derkach [3], Binbin Zheng-Lin[1], Venkata Yellapantula[1], Benjamin Diamond [1,4], Malin Hultcrantz [1], Bachisio Ziccheddu[4,5], Eileen M. Boyle[6], Patrick Blaney[6], Niccolò Bolli[7,8], Yanming Zhang[9], Ahmet Dogan [10], Alexander M. Lesokhin[1,11], Gareth J. Morgan [6], Ola Landgren [4,12] & Francesco Maura [4,12✉]

Chromothripsis is detectable in 20–30% of newly diagnosed multiple myeloma (NDMM) patients and is emerging as a new independent adverse prognostic factor. In this study we interrogate 752 NDMM patients using whole genome sequencing (WGS) to investigate the relationship of copy number (CN) signatures to chromothripsis and show they are highly associated. CN signatures are highly predictive of the presence of chromothripsis (AUC = 0.90) and can be used identify its adverse prognostic impact. The ability of CN signatures to predict the presence of chromothripsis is confirmed in a validation series of WGS comprised of 235 hematological cancers (AUC = 0.97) and an independent series of 34 NDMM (AUC = 0.87). We show that CN signatures can also be derived from whole exome data (WES) and using 677 cases from the same series of NDMM, we are able to predict both the presence of chromothripsis (AUC = 0.82) and its adverse prognostic impact. CN signatures constitute a flexible tool to identify the presence of chromothripsis and is applicable to WES and WGS data.

[1] Myeloma Service, Department of Medicine, Memorial Sloan Kettering Cancer Center, New York, NY, USA. [2] Institute for Cancer Research, Oslo University Hospital Radiumhospitalet, Oslo, Norway. [3] Department of Epidemiology and Biostatistics, Memorial Sloan Kettering Cancer Center, New York, NY, USA. [4] Myeloma Service, Sylvester Comprehensive Cancer Center, University of Miami, Miami, FL, USA. [5] Department of Molecular Biotechnologies and Health Sciences, University of Turin, Turin, Italy. [6] Myeloma Research Program, NYU Langone, Perlmutter Cancer Center, New York, NY, USA. [7] Department of Oncology and Hemato-Oncology, University of Milan, Milan, Italy. [8] Hematology Unit, Fondazione IRCCS Ca' Granda Ospedale Maggiore Policlinico, Milan, Italy. [9] Cytogenetics Laboratory, Department of Pathology, Memorial Sloan Kettering Cancer Center, New York, NY, USA. [10] Hematopathology Service, Department of Pathology, Memorial Sloan Kettering Cancer Center, New York, NY, USA. [11] Department of Medicine, Weill Cornell Medical College, New York, NY, USA. [12] These authors jointly supervised this work: Ola Landgren, Francesco Maura. ✉email: fxm557@med.miami.edu

Chromothripsis, a catastrophic chromosomal shattering event associated with random rejoining, is emerging as strong and independent prognostic factor across multiple malignancies[1–7]. Reliable detection of chromothripsis requires whole-genome sequencing (WGS) and the integration of both structural variants (SVs) and copy number (CN) data, with manual inspection remaining the gold standard approach[1,2,4,8].

Recently, we reported a comprehensive study of structural variation (SV) in a series of 752 newly diagnosed multiple myeloma (NDMM) from the CoMMpass trial for which long-insert low-coverage WGS was available (NCT01454297)[9]. Using the latest criteria for chromothripsis[1–5] and manual inspection, we reported a 24% prevalence of chromothripsis, making multiple myeloma (MM) the hematological cancer with the highest documented prevalence of chromothripsis[1,5,10,11]. MM patients with chromothripsis events were characterized by poor clinical outcomes, with chromothripsis being associated with multiple unfavorable clinical and genomic prognostic factors including translocations involving *MAF*, *MAFB*, and *MMSET*, increased APOBEC mutational activity, del17p13 and *TP53* mutations[9].

Chromothripsis can be detected in myeloma precursor conditions (monoclonal gammopathy of undetermined significance, MGUS and smoldering myeloma, SMM) years before progression, representing an early genomic MM-defining event which is highly predictive for later progressive disease[12,13]. In MM, the molecular time of major gains induced by chromothripsis suggests a significant fraction of these complex events are acquired decades before diagnosis[14]. As further confirmation of early acquisition and its driver role, the structure of chromothripsis tends to be stable during spontaneous progression from precursor conditions to MM, and equally is conserved at post-therapy relapse[12,14,15] (Supplementary Fig. 1). In contrast, chromothripsis in solid cancer is often noted to be a late event, arising in metastatic disease or post-therapy samples, with a structure that can be unstable over time[1,16,17].

In MM and in other hematological malignancies, the structural complexity of each chromothripsis event is typically more simple than that seen in solid cancers[1,4,5] (Supplementary Fig. 2). Specifically, the total focal CN gains within the regions of chromothripsis is often lower than in solid organ cancer, there are fewer breakpoints attributable to chromothripsis, and in MM there is a lack of enrichment for double-minutes and other more catastrophic events such as *typhonas*[1,8].

SV and CN signatures have been reported in ovarian cancer as potential BRCAness surrogates[18,19]. This important marker, denoting both prognosis and treatment-responsiveness, is detectable only by combining multiple WGS features[18,19]. Of note, CN signatures alone are an independent predictor of clinical outcomes from ovarian cancer WGS[18]. Given the genome-wide distribution and lower complexity of chromothripsis in MM, we hypothesized that a comprehensive signature analysis approach using SV and CN may provide an accurate estimation of chromothripsis in MM.

Here we demonstrate, by using the NDMM CoMMpass trial low-coverage long-insert WGS ($n = 752$), that the combination of CN with SV signatures is equivalent to a previously described chromothripsis-calling algorithm (i.e., *ShatterSeek*)[1]. Interestingly, CN signatures alone remain highly predictive of chromothripsis, without requiring specific SV assessment, demonstrated in the CoMMpass data as well as an additional validation set of WGS from NDMM ($n = 34$) and other hematological malignancies ($n = 235$). Furthermore, in the CoMMpass dataset, we show that CN signatures independently associate with shorter progression-free (PFS) and overall survival (OS). Leveraging the ability of CN signatures to predict chromothripsis without SV data, we extend the analysis to whole-exome sequencing (WES),

where we confirm the ability of CN signatures to predict the presence of chromothripsis and demonstrate that WES-based CN signatures retain the association with adverse clinical outcomes. Utilizing comprehensive signature assessment in WES data potentially accelerates the clinical translation of testing for chromothripsis where WGS data is not available.

## Results

**Experimental data and design.** Genome-wide somatic CN profiles were generated from 752 NDMM patients with low-coverage long-insert WGS (median 4–8×) from the CoMMpass study (NCT01454297; IA13; Supplementary Table 1)[20,21]. The final SV catalog was generated by combining the two SV calling algorithms, DELLY[22], and Manta[23] with CN data, followed by a series of quality filters (see "Methods")[9]. According to the most recently published criteria[1–5], at least one chromothripsis event was observed in 24% of the entire series[9].

**De novo CN signature extraction in multiple myeloma.** CN signature analysis takes the genome-wide CN gains and losses (Fig. 1a), and measures 6 fundamental CN features: (i) number of breakpoints per 10 Mb, (ii) absolute CN of segments, (iii) difference in CN between adjacent segments, (iv) number of breakpoints per chromosome arm, (v) lengths of oscillating CN segment chains, and (vi) the size of segments (Fig. 1b)[18]. The optimal number of categories in each CN feature was established using a mixed effect model with the *mclust* R package (Fig. 1c, d). The consequence of taking this approach is that different malignancies and types of sequencing data may result in varying numbers of CN categories and thresholds defining these categories (see "Methods")[18].

To take account of the biology of MM, we introduced a few modifications to the original CN features described by Macintyre et. al.:[18] (i) given the known poor quality mapping and copy number complexity related to class switch recombination and VDJ rearrangements, the immunoglobulin regions corresponding to IgH, IgL, and IgK were removed; (ii) considering both the low-coverage long-insert WGS limitation for calling subclonal copy number events and the less complex MM karyotype compared to solid cancers, fixed criteria for copy number status were introduced (see "Methods", and Supplementary Data 1 for full analytical R code).

Analyzing the CoMMpass long-insert low-coverage WGS; 28 CN categories were defined (Fig. 1d; Supplementary Table 2). In comparison to the CN features described in ovarian cancer by shallow WGS[18], in MM we observe lower total CN (median; 2, maximum; 9, compared with total CN exceeding 30 in a proportion of ovarian cancer). These findings are supported by the Pan-Cancer Analysis of Whole Genomes (PCAWG) dataset, where within each patient harboring chromothripsis, those with ovarian cancer had a higher number of chromosomal breakpoints [median 62, interquartile range [interquartile range (IQR) 33–112] compared with median 24 (IQR 12–35) for full coverage MM, and median 25 (IQR 15–43) for low coverage MM, each $p < 0.0001$, Supplementary Fig. 2]. We also note shorter lengths of oscillating CN, and a low contribution from very large aberrant segments (in comparison to the dominant contribution from segments >30 Mb in ovarian CN signature #1)[18]. Overall, these differences are in line with the lower genomic complexity of MM compared to ovarian cancer and the majority of solid organ cancers (Supplementary Fig. 2).

Running the hierarchical Dirichlet process (*hdp*), 5 CN signatures were extracted in MM (Fig. 2; Supplementary Table 3). CN-SIG1, CN-SIG2, and CN-SIG3 have high contributions from CN categories representing low numbers of breakpoints per

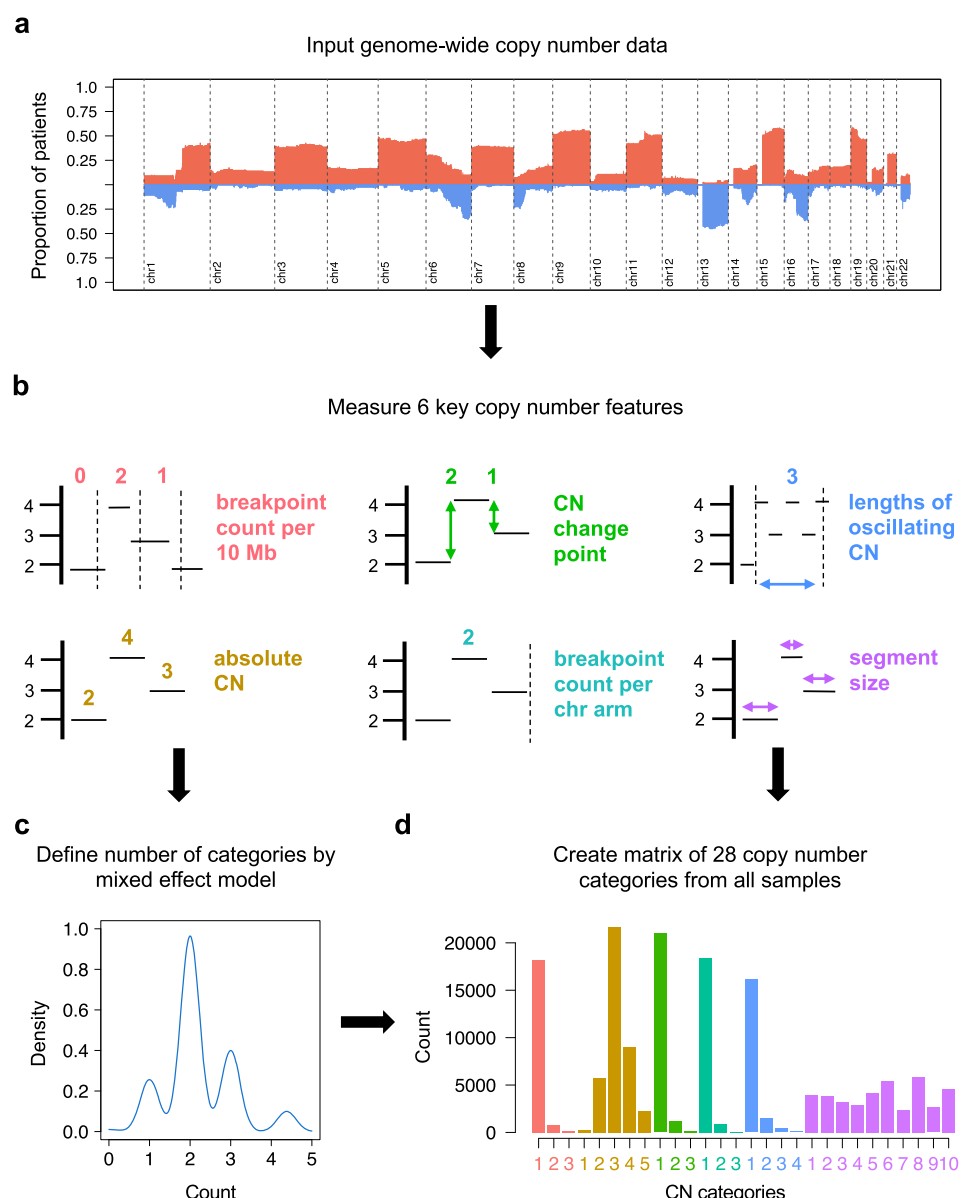

**Fig. 1 A schema demonstrating the definition of copy-number (CN) features from multiple myeloma whole-genome sequencing data. a** Input genome-wide copy number gain (red) and loss (blue) data from 752 newly diagnosed multiple myeloma whole genomes. **b** Measure copy number as classified by 6 key features. **c** Define the optimum number of categories for each copy number feature by a mixed-effects model (*mclust*). **d** Tally the number of CN variation for each of 28 CN categories to produce a matrix of key CN features. This comprises the input matrix for the hierarchical Dirichlet process (*hdp*) for de novo extraction of CN signatures.

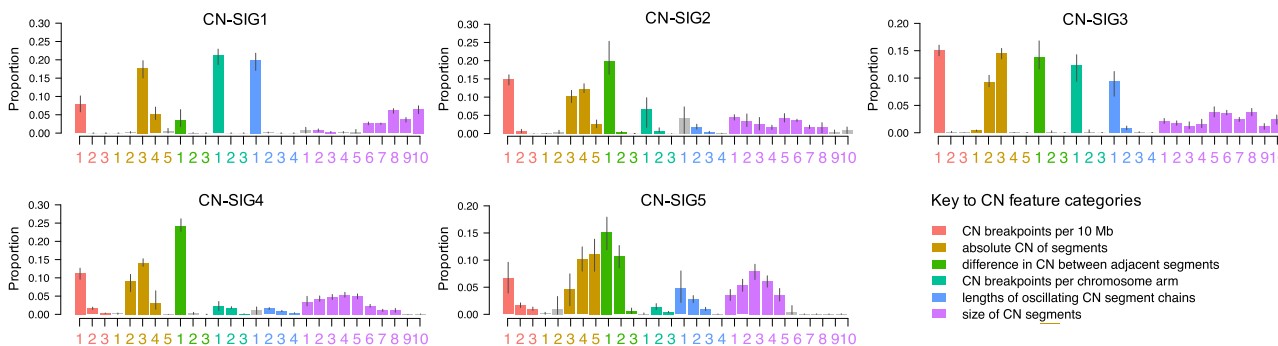

**Fig. 2 De novo extraction from whole-genome sequencing data produces 5 copy-number (CN) signatures in 752 newly diagnosed multiple myeloma.** The 5 CN signatures extracted comprise varying contribution across the 28-CN-feature matrix. The 2 chromothripsis-associated signatures are CN-SIG4 and CN-SIG5. (CN-SIG: copy-number signature, *n* = 752 samples, data are presented as median values ± SD).

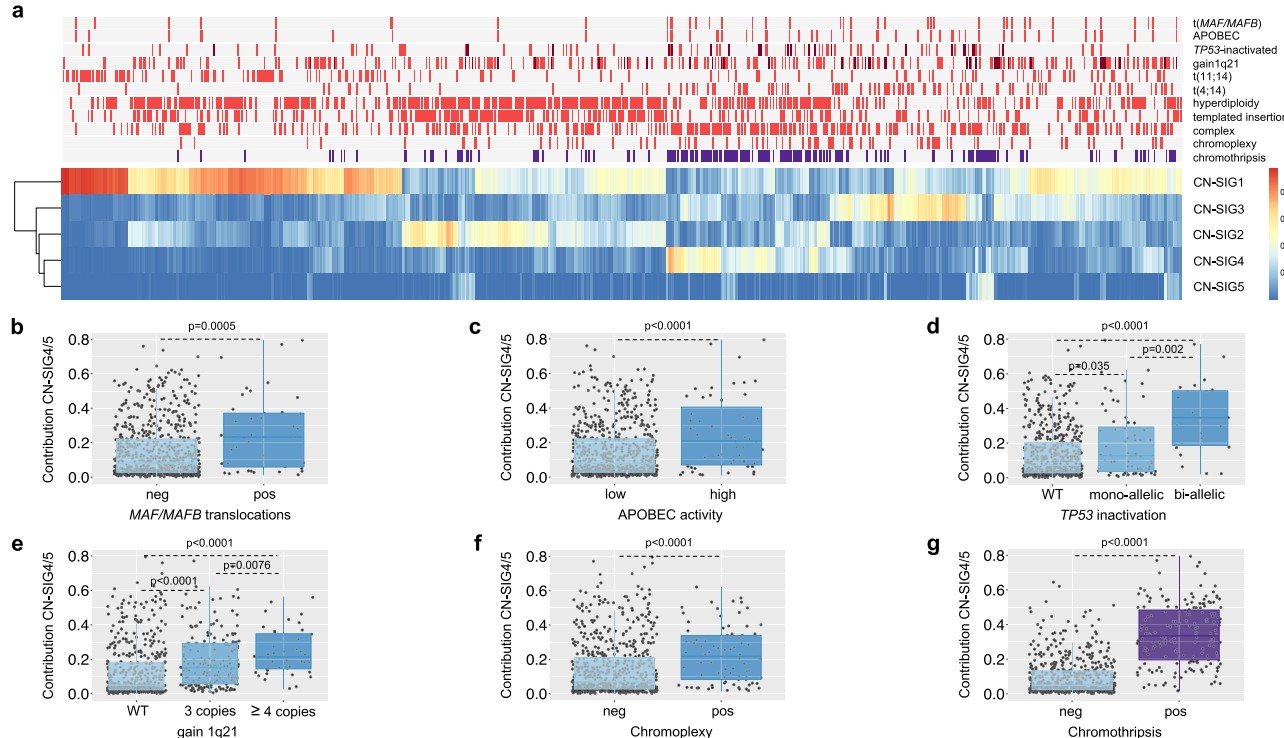

**Fig. 3 Clinical data demonstrates the correlation of copy number (CN) signatures with high-risk multiple myeloma prognostic features and complex genomic change. a** A heatmap of MM mutational and structural features demonstrates that contribution from CN-SIG4 and CN-SIG5 cluster with features of high-risk MM. Presence of biallelic *TP53* inactivation and chromosome 1q21 amplification (i.e., >3 copies) are annotated in dark red; presence of chromothripsis in purple; all the other genomic features are in bright red when present. **b–g** There is a significantly higher median contribution from CN-SIG4 and/or CN-SIG5 on the samples having translocations involving **b** *MAF/MAFB* ($p = 0.0005$), **c** increased APOBEC mutational activity ($p = 9.9e^{-5}$), **d** biallelic *TP53* inactivation ($p = 1.3e^{-6}$), **e** gain/amplification of chromosome 1q21 ($p = 1.3e^{-8}$), **f** chromoplexy ($p = 1.7e^{-7}$) and **g** chromothripsis ($p = 2.2e^{-16}$). Boxplots show median and interquartile range (IQR), with whiskers extending to 1.5 * IQR, $n = 752$, $p$-values indicate significance by a 2-sided Wilcoxon rank-sum test. (CN-SIG: copy number signature, neg: lacking the feature, pos: containing the feature, WT: wild type).

10 Mb and breakpoints per chromosome arm. These signatures have small absolute differences between adjacent CN segments and short lengths of oscillating copy number. Each signature varies in the distribution of segment size and in the relative contribution of each CN category; CN-SIG1 has minimal jumps between adjacent segments and a higher contribution from larger segment sizes, mostly single chromosomal gains and trisomies. CN-SIG2 has higher total CN (i.e., multiple chromosomal gains and tetrasomies) and a higher contribution from small segments without jumps between adjacent segments; and CN-SIG3 is enriched for low absolute CN (i.e., deletions) with usually isolated events (rare oscillating events or multiple events on the same arm/chromosome) (Fig. 2). In contrast, CN-SIG4 and CN-SIG5 were characterized by higher numbers of breakpoints per 10 Mb and per chromosome arm, longer lengths of oscillating CN, and a higher contribution from small segments of CN change. While, CN-SIG4 has contribution from each of the 3 categories reflecting longer oscillation lengths, CN-SIG5 was characterized by a higher contribution from jumps in CN between adjacent segments, and by a higher contribution from high absolute CN (Fig. 2).

**CN signatures across multiple myeloma defining genomic events.** Given the complex CN features noted in CN-SIG4 and CN-SIG5 (Fig. 2), we examined the association of these signatures with known MM genomic features (i.e., MM defining genomic events)[9,12,14,20,24–26]. Both signatures were correlated with features of high-risk MM (Fig. 3a), including translocations involving *MAF/MAFB* ($p = 0.0005$), APOBEC mutational activity (i.e.,

mutational signatures; $p < 0.0001$), biallelic *TP53* inactivation ($p < 0.0001$), and gain/amplification of 1q21 ($p < 0.0001$; Fig. 3b-e). There was a negative association with t(11;14)(*CCDN1;IGH*) ($p < 0.0001$; Supplementary Fig. 3a), consistent with the low genomic complexity known to be associated with a large proportion of this molecular subgroup of MM[14,20]. We show that CN-SIG4 and CN-SIG5 are highly correlated with the presence of complex structural chromosomal rearrangements (Fig. 3a), including the subgroup of chromoplexy ($p < 0.0001$; Fig. 3f), and rearrangements defined as "complex- not otherwise specified" (complex-NOS; $p < 0.0001$; Supplementary Fig. 3b; "Methods"). Interestingly, the largest significant difference was noted with chromothripsis; with a median contribution of CN-SIG4/5 of 0.33 being seen in those cases with chromothripsis (IQR 0.20–0.48) compared with 0.05 in those without (IQR 0.02–0.13) ($p < 0.0001$; Fig. 3g). These data suggest that CN signature analysis has the potential to accurately predict the presence of chromothripsis from WGS data derived from MM patients.

**CN signatures are strongly predictive of chromothripsis in multiple myeloma.** Considering that previously described chromothripsis detection methods utilize both CN and SV from WGS, we tested the accuracy of prediction by CN signatures combined with SV signatures, previously described in solid organ cancer[4,19] and MM[27]. Running *hdp*, 10 SV signatures were extracted in MM ("Methods", Supplementary Data 2, Supplementary Table 4), with 3 comprising clustered events consistent with chromothripsis (SV-SIG1-3) and 7 comprising non-clustered events (Fig. 4).

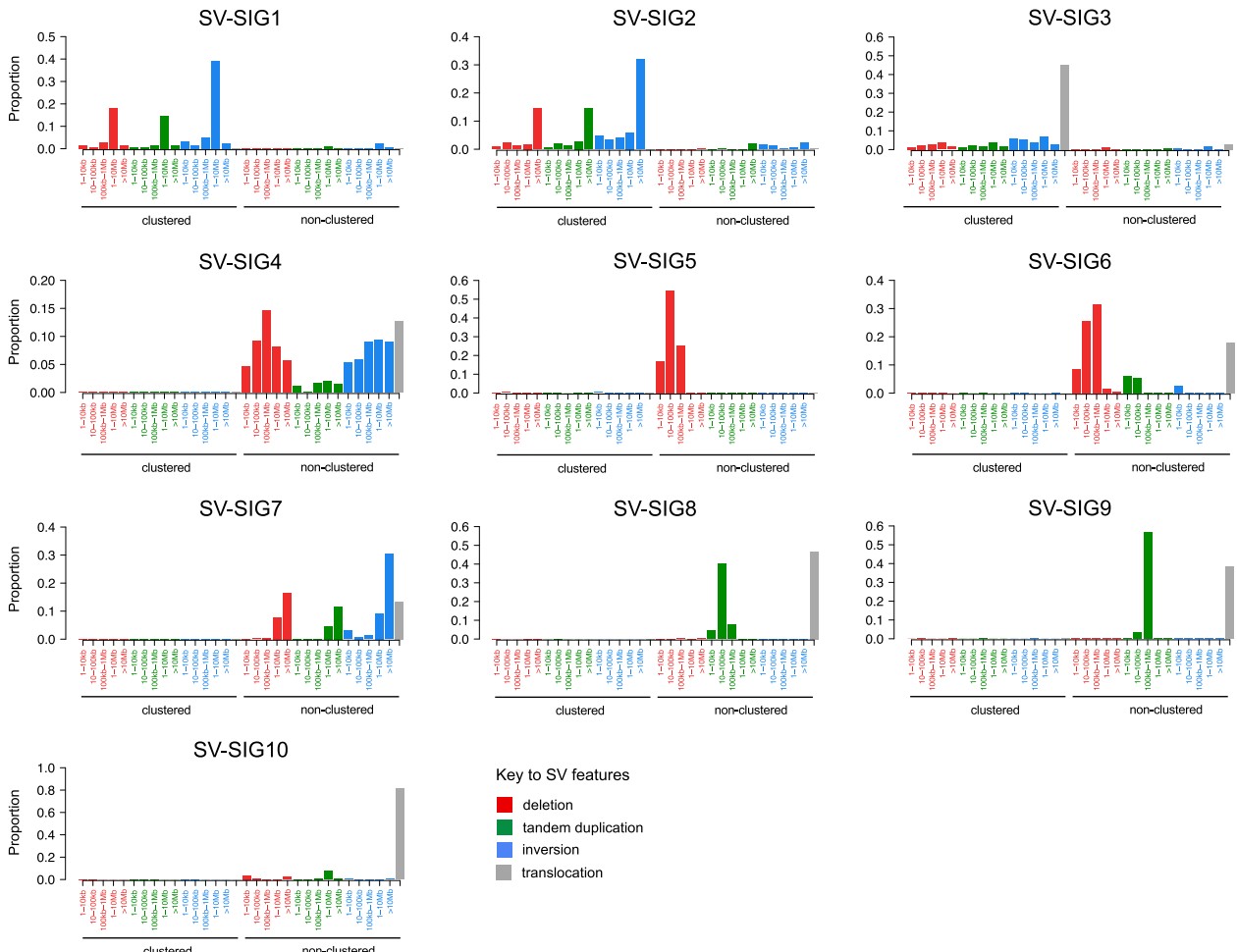

**Fig. 4 De novo extraction from whole-genome sequencing data produces 10 structural variant (SV) signatures in 752 newly diagnosed multiple myeloma samples.** The 10 SV signatures extracted comprise varying contribution across the 32-SV-feature matrix. SV-SIG1, SV- SIG2, and SV-SIG3 contain clustered SV features and are associated with chromothripsis. The remainder of the signatures consist of non-clustered events. (SV-SIG: structural variant signature).

SV-SIG1-3 correlate with the presence of complex structural chromosomal rearrangements, and cluster together with CN-SIG4 and CN-SIG5 (Supplementary Fig. 4). Evaluation of prediction accuracy of CN and SV signatures by receiver operating curve (ROC) analysis with 10-fold cross validation shows that genomic signatures are highly predictive of chromothripsis; producing an average area-under-the-curve (AUC) of 0.96 (Fig. 5a; for the full analytical R code, see Supplementary Data 3). This prediction is as accurate as that obtained using *ShatterSeek* (AUC = 0.93, Fig. 5b) and with SV signatures alone (AUC = 0.94, Fig. 5c). Importantly, we demonstrate that CN signatures alone retains a highly accurate prediction (AUC = 0.90, Fig. 5d, Supplementary Table 5), statistically non-inferior to *ShatterSeek* (AUC difference = 0.03, standard deviation[SD] = 0.016, *p* = 0.54, based on bootstrap analysis), without the requirement for specific SV evaluation, potentially allowing the detection of chromothripsis on non-WGS. Given the higher proportional contribution of CN-SIG4, the prediction of chromothripsis is driven by this signature, however, CN-SIG5 contribution adds to the prediction of chromothripsis, with the AUC from CN-SIG4 alone being 0.88 (AUC difference = 0.03, standard deviation[SD] = 0.01, *p* = 0.009, based on bootstrap analysis).

Due to the association of CN-SIG4/5 with chromoplexy and other complex SV (Fig. 3), we tested the accuracy of predicting these events by CN signatures, confirming that the prediction was more specific for chromothripsis (AUC = 0.74 for each of chromoplexy and complex SV, Supplementary Fig. 5). To confirm that CN-SIG4/5 predict chromothripsis events and not general patterns of genomic complexity, we observed that the both the CN and SV profiles outside of chromothripsis were highly analogous with samples lacking chromothripsis, (cosine similarity for corresponding matrix columns; CN 0.98, SV 0.96, Supplementary Figs. 6–7), suggesting the absence of a complex and unstable background genome. This is in line with the notation that chromothripsis in NDMM often represents the most complex event occurring, rarely associated with the hallmarks of genomic instability observed in solid cancer[28,29].

Next, we examined discrepant cases, defined as "false positive" when chromothripsis was absent on manual curation but predicted by CN signatures, or "false negative" when chromothripsis was present but not predicted by CN signatures. Here we found that the structural complexity of chromothripsis contributed to its detection by CN signatures (Supplementary Fig. 6). In the false positive cases, 17/23 (74%) contained complex SV (not otherwise classified), some of which represent chromothripsis-like events occurring without the minimum of 10 SV required for classification[2,5,9]. In contrast, restricting analysis to chromothripsis cases having more chromosomal breakpoints increases the AUC of prediction [AUC = 0.93 with >30 breakpoints (71/752), AUC = 0.96 with >50 breakpoints (37/752)]. Taken all together, these data suggest that the prediction

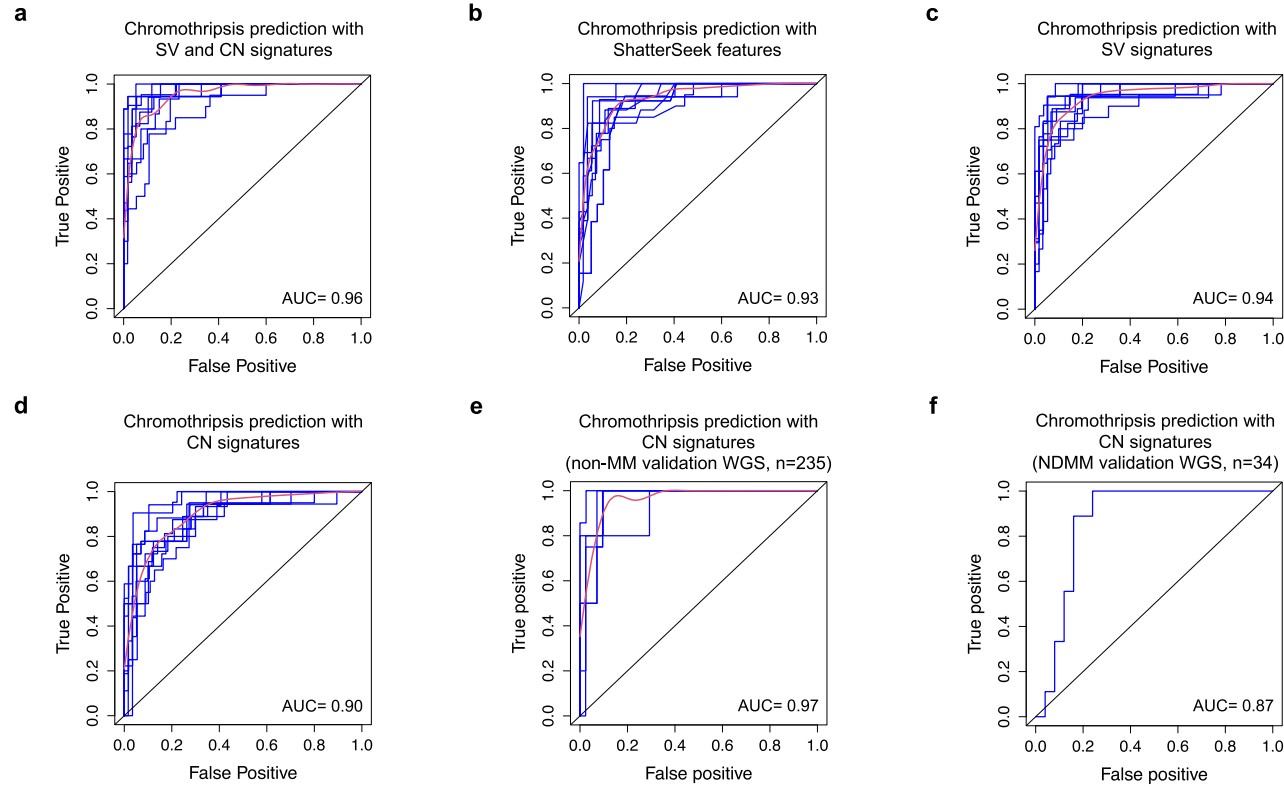

**Fig. 5 Copy number (CN) signatures in newly diagnosed multiple myeloma are strongly predictive of chromothripsis.** Receiver operating curve (ROC) for the prediction of chromothripsis from CoMMpass whole-genome sequencing (WGS) data ($n = 752$) from **a** structural variant (SV) and CN signature analysis, **b** *ShatterSeek* features, **c** SV signatures alone and **d** CN signatures alone. **e** ROC for the prediction of chromothripsis from the validation set of other hematological cancers ($n = 235$). **f** ROC for the prediction of chromothripsis from the newly diagnosed multiple myeloma subset of the validation WGS ($n = 34$). Blue lines represent individual ROC (from 10-fold cross validation in (**a**–**e**) and 5-fold validation in (**f**), red lines represent the mean of individual ROC, AUC: mean area-under-the-curve.

of chromothripsis by CN signatures is most accurate in cases with higher complexity.

**CN signatures are strongly predictive of chromothripsis in hematological malignancies.** Given the low documented prevalence and low complexity of chromothripsis in hematological cancers[1,10,11], we validated our prediction model using an extended dataset of 269 full coverage WGS from previously published hematological cancer samples, including data from the PCAWG study ($n = 269$)[5,29,30]. This included 34 NDMM, 92 chronic lymphocytic leukemia, 29 chronic myeloid leukemia, 104 B-cell lymphoma and 10 acute myeloid leukemia (7 de novo, 3 therapy-related) (Supplementary Table 6; "Methods"). Overall, the number of categories extracted in this series of WGS was smaller compared to the CoMMpass cohort (26 vs 28), likely reflecting the less impaired cytogenetic profile of most non-MM hematological cancers[4]. Following the same computational approach reported in Supplementary Data 1 and 2, de novo extraction on the entire validation cohort identified 4 CN-signatures that were highly similar to those described in the CoMMpass WGS (Supplementary Fig. 8; Supplementary Tables 7–8). Across the cohort of non-MM hematological malignancies, ($n = 235$), the resultant ROC analysis had an AUC of 0.97 for predicting chromothripsis (Fig. 5e, using 5-fold cross validation due to the smaller sample size), while an AUC of 0.87 was observed when testing only in NDMM ($n = 34$) (Fig. 5f). These data demonstrate the reproducibility of chromothripsis prediction from CN signatures, in both a separate set of hematological cancer WGS, and an independent set of NDMM samples.

**CN signatures are strongly predictive of clinical outcomes in multiple myeloma.** Survival analysis on the CoMMpass data demonstrates that the presence of chromothripsis is one of the strongest predictors of a shorter PFS and OS;[9] median PFS of 32.2 months (95% confidence interval [CI] 25.2–48.3 m) in those harboring chromothripsis compared with 41.1 m (95% CI 37.8–47.2 m) in those without ($p = 0.00011$; Supplementary Fig. 9a), and median OS of 53.3 m with chromothripsis but not reached [NR] in those without ($p < 0.0001$; Supplementary Fig. 9b). Survival probability according to the CN-signature predictive model mirrored survival according to chromothripsis. Those with a high CN_pred score, defined as a predicted chromothripsis probability ≥0.6 (see "Methods"), had a median PFS of 29.7 m (95% CI 25.2m-NR) compared with 41.8 m (95% CI 38.0–48.1 m) in those with a low score ($p = 0.0017$; Fig. 6a; Supplementary Table 5). Median OS in those with high CN_pred score was also significantly shorter at 53.1 m compared with NR in those with a low score ($p < 0.0001$; Fig. 6b).

To select the most important features from highly correlated genomic risk factors (Fig. 3a) we performed a backwards stepwise Cox regression including ISS, age, ECOG status, biallelic *TP53* inactivation, t(4;14)(*FGFR3;IGH*), gain/amplification1q21, increased APOBEC mutational activity and *MAF/MAFB* translocations. Based on this approach the final model contained ISS, age, ECOG, APOBEC mutational activity, gain/amplification1q21 and the CN_pred score. The model is consistent with previously published data indicating that APOBEC mutational activity is one of the strongest adverse prognostic factors in MM[24,26,31], and that gain/amplification1q21 is associated with early relapse[32]. The CN_pred score showed a significant association with shorter PFS

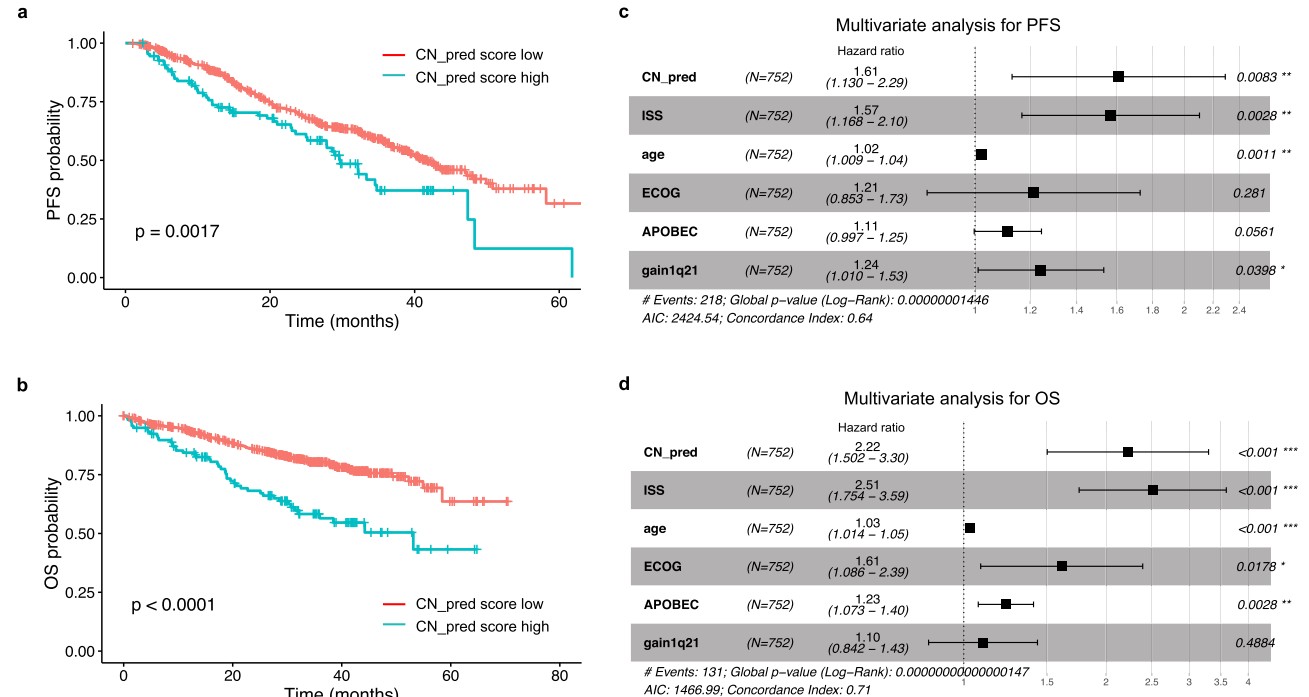

**Fig. 6 Copy number (CN) signatures in newly diagnosed multiple myeloma are independently predictive of clinical outcomes. a** Progression-free survival (PFS) probability in the CoMMpass dataset according to high (blue) or low (red) CN-prediction score for chromothripsis (CN_pred). **b** Overall survival (OS) probability in the CoMMpass dataset according to high (blue) or low (red) CN_pred. **c** Multivariate analysis of the effect of CN_pred on PFS after correction for International Staging Score (ISS), age, Eastern Cooperative Oncology Group (ECOG) score, and APOBEC mutational activity. **d** Multivariate analysis of the effect of CN_pred on OS after correction for the same factors. All p-values for Kaplan–Meier curves were generated according to a 2-sided log-rank test. Multivariate analysis was performed by the Cox proportional hazards model with p-values according to a 2-sided Wald test. Data are presented as median values ± 95% confidence interval.

and OS after controlling for other variables in the model, producing a hazard ratio (HR) of 1.61 (95% CI 1.13–2.29, $p = 0.0083$, Fig. 6c), and 2.22 (95% CI 1.50–3.30, $p < 0.001$, Fig. 6d), respectively. Importantly, the association of the CN_pred score with poor clinical outcome was independent of treatment received (Supplementary Fig. 10).

When examining false positive and false negative cases, multivariate analysis including the same risk factors as Fig. 6 demonstrated no significant effect on survival in discrepant cases compared with negative cases (Supplementary Fig. 11). In comparison, cases in which chromothripsis was present and had been predicted with CN signatures showed significantly shorter survival, (PFS HR 1.73, 95% CI 1.16–2.58, $p = 0.0068$; OS HR 2.38, 95% CI 1.54–3.67, $p < 0.001$). Taken together this suggests that the poor prognosis associated with chromothripsis is mostly driven by highly complex events.

**CN signatures compared with other CN-based tools.** We next compared the prediction of the presence of chromothripsis by CN signatures with other CN-based algorithms recently used in MM to identify high-risk disease: a loss-of-heterozygosity index (LOH_index)[20] and the genomic scar score (GSS)[31,33] (see "Methods"). Results from each of these CN assessment approaches showed a right-skewed distribution of CN features, [LOH_index; median 2, (range 0–27), Supplementary Fig. 12a, and GSS; median 7, (range 0–39), Supplementary Fig. 12b], with the GSS distribution closely resembling that of previously published data in NDMM[31].

Each of these approaches demonstrated a lower average AUC for predicting the presence of chromothripsis in MM WGS (0.69 and 0.78 respectively, Supplementary Fig. 12c, d). The difference

in chromothripsis prediction between CN signatures and the LOH_index is quantitated as a statistically significant difference of 0.21 in AUC (based on bootstrap analysis, SD = 0.006, $p < 0.0001$, Supplementary Fig. 12e) while the difference in prediction between CN signatures and the GSS is quantitated as a statistically significant difference of 0.13 in AUC (based on bootstrap analysis, SD = 0.005, $p < 0.0001$, Supplementary Fig. 12f, Supplementary Data 4).

In order to compare the effect on PFS and OS in multivariate analysis, the CN-signature prediction data was used as a linear variable, which after correction for the previously included risk factors (see Fig. 6c, d) was associated with shorter PFS (HR = 1.87, 95% CI 1.16–3.01, $p = 0.012$, Supplementary Fig. 13a) and OS (HR = 3.1, 95% CI 1.84–5.4, $p < 0.001$, Supplementary Fig. 13d). Performing multivariate analysis for PFS with correction for the same risk factors showed that neither the LOH_index (PFS HR = 1.03, 95% CI 0.99–1.08, $p = 0.19$; Supplementary Fig. 13b) nor the GSS (PFS HR = 1.02, 95% CI 1.0–1.04, $p = 0.12$; Supplementary Fig. 13c) retain a significant association. Each model has a slightly increased HR for OS in multivariate analysis; (LOH_index HR = 1.1, 95% CI 1.02–1.1, $p = 0.008$, Supplementary Fig. 13e; GSS HR = 1.0, 95% CI 1.02–1.1, $p = 0.001$, Supplementary Fig. 13f). Overall, CN-signatures perform significantly better at predicting poor outcomes in comparison with either the LOH_index or the GSS, suggesting that a more accurate prediction of chromothripsis is a better tool for identifying prognosis using CN-based information.

**CN signatures predict chromothripsis and clinical outcomes in whole-exome sequencing data.** Any prognostic assessment for MM would ideally be applicable in non-WGS assays, as WGS is

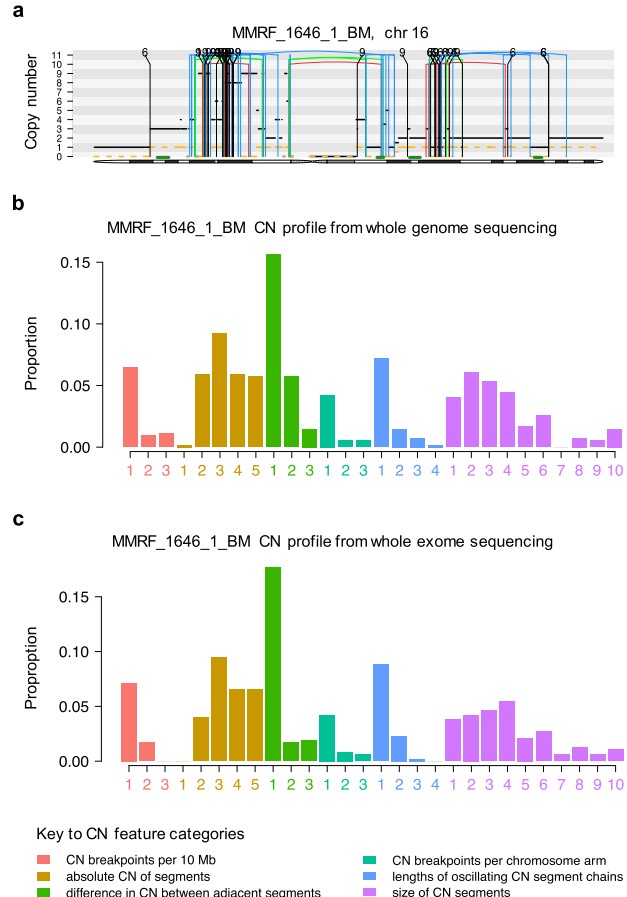

**a**

MMRF_1646_1_BM, chr 16

**b**

MMRF_1646_1_BM CN profile from whole genome sequencing

**c**

MMRF_1646_1_BM CN profile from whole exome sequencing

Key to CN feature categories

- CN breakpoints per 10 Mb
- absolute CN of segments
- difference in CN between adjacent segments
- CN breakpoints per chromosome arm
- lengths of oscillating CN segment chains
- size of CN segments

**Fig. 7 Extracted copy number (CN) feature profiles from whole-genome sequencing (WGS) and whole-exome sequencing (WES) are highly analogous. a** An example of chromothripsis from the CoMMpass dataset (MMRF_1646_1_BM; chr: chromosome). The horizontal black line indicates total copy number; the dashed orange line minor copy number. Vertical lines represent structural variant breakpoints for deletion (red), inversion (blue), tandem-duplication (green), and translocations (black), involving chromosomes 6 and 9. The extracted CN category profile from the same example patient (MMRF_1646_1_BM) from **b** WGS and **c** WES.

currently both expensive and computationally intensive, making its clinical application outside of a research setting difficult. We performed de novo signature extraction using WES data from 677 NDMM CoMMpass samples, all of which also had WGS. The presence of these data enabled us to compare results in WES with the gold-standard method for chromothripsis-detection on WGS. The CN feature profile extracted from WES data was highly analogous to that obtained from WGS data (cosine similarity = 0.99 for corresponding matrix columns), with a smaller contribution from the oscillation CN categories due to the lower data resolution overall, and in particular of focal and small lesions (Fig. 7, Supplementary Fig. 14a). De novo extraction using *hdp* produced 5 exome-based CN signatures (eCN), similar in their CN feature distribution to the signatures defined in WGS (Supplementary Figs. 14b, c; Supplementary Tables 8–9). ROC analysis based on 10-fold validation produced an average AUC of 0.82 for predicting chromothripsis (Supplementary Fig. 15; Supplementary Table 5).

The exome CN signature-based chromothripsis prediction score (eCN_pred) was associated with a significantly shorter PFS; median 26.0 m (95% CI 18.0–48.3 m) in those with a high eCN_pred score compared with 41.1 m (95% CI 36.7–50.0 m) in

those with a low score, ($p = 0.0031$; Fig. 8a). OS was also significantly shorter; median 52.3 m with a high eCN_pred score but NR in those with a low score, ($p < 0.0001$; Fig. 8b).

In the WES data, backwards stepwise regression demonstrated that the best model for predicting survival was that comprising age, ISS, APOBEC-activity, and the eCN_pred score. Multivariate analysis again produced a significant and independent association of eCN-pred with a shorter PFS (HR = 1.66, 95% CI 1.16–2.37, $p = 0.0055$; Fig. 8c), and shorter OS (HR = 2.19, 95% CI 1.46–3.29, $p < 0.001$; Fig. 8d) recapitulating both the results obtained from WGS CN signature-based chromothripsis prediction (Fig. 6), and those obtained by manual data curation[9].

## Discussion

We recently carried out a comprehensive analysis of the landscape of SVs in MM, showing their critical role in disease pathogenesis and confirming the importance of WGS for deciphering the genomic complexity of these events[9,14,34]. We demonstrated a high prevalence of complex structural events such as chromothripsis in MM (24%). In contrast to solid cancers having a similar or higher prevalence, in MM chromothripsis represents an early driver event detectable years before the diagnosis, which remains relatively stable over time. Importantly, chromothripsis is emerging as one of the strongest features able to predict both the progression of myeloma precursor condition to MM and shorter PFS and OS with NDMM, independent of other known prognostic variables[7,9,12]. Given these relevant translational and clinical data in MM and other malignancies[7,9,35,36], it follows that the integration of complex SV data has the potential to improve the current prognostic scoring systems.

Current approaches for identifying chromothripsis require expense and time commitment because of the need for either the manual curation of WGS data or the use of computational tools requiring both CN and SV data[1,8,9,37]. Genomic signatures provide a comprehensive assessment of multi-dimensional data to predict the presence of complex SV. A signature approach has already been applied to breast and ovarian cancer, where BRCA deficiency can be predicted by distinct SV and CN features[18,19,38].

We investigated using a genomic signature approach for predicting the presence of chromothripsis in NDMM, demonstrating that SV and CN signatures accurately predict chromothripsis, with a performance analogous to *ShatterSeek*. CN signature analysis alone provides a highly accurate chromothripsis prediction, independent from SV signatures and outperforming other CN assessment algorithms[31]. In comparison to solid cancers, in MM the genomic background on which chromothripsis occurs is not complex or unstable. This is expected considering the relative absence of patterns of genomic instability and the early driver role of chromothripsis in MM, while the acquisition of other known features responsible for genomic complexity such as *TP53*-inactivation and APOBEC- mutational activity occur later[26,39]. The survival probability identified using a CN signature-based prediction of chromothripsis closely mimics PFS and OS curves observed with the presence/absence of chromothripsis[9]. Using a validation set of WGS containing multiple hematological malignancies, we provide proof-of-principle that CN signature analysis can predict for chromothripsis across different hematological cancer types and can, therefore, be used as surrogate for these variants to further address the role of chromothripsis in these blood cancers.

The primary objective was to test whether WGS-based CN signatures can reliably predict chromothripsis and its poor impact on clinical outcomes, independently from other WGS data. Another critical aspect of this study was to expand our investigations using non-WGS (i.e., exome-based) data. In WES

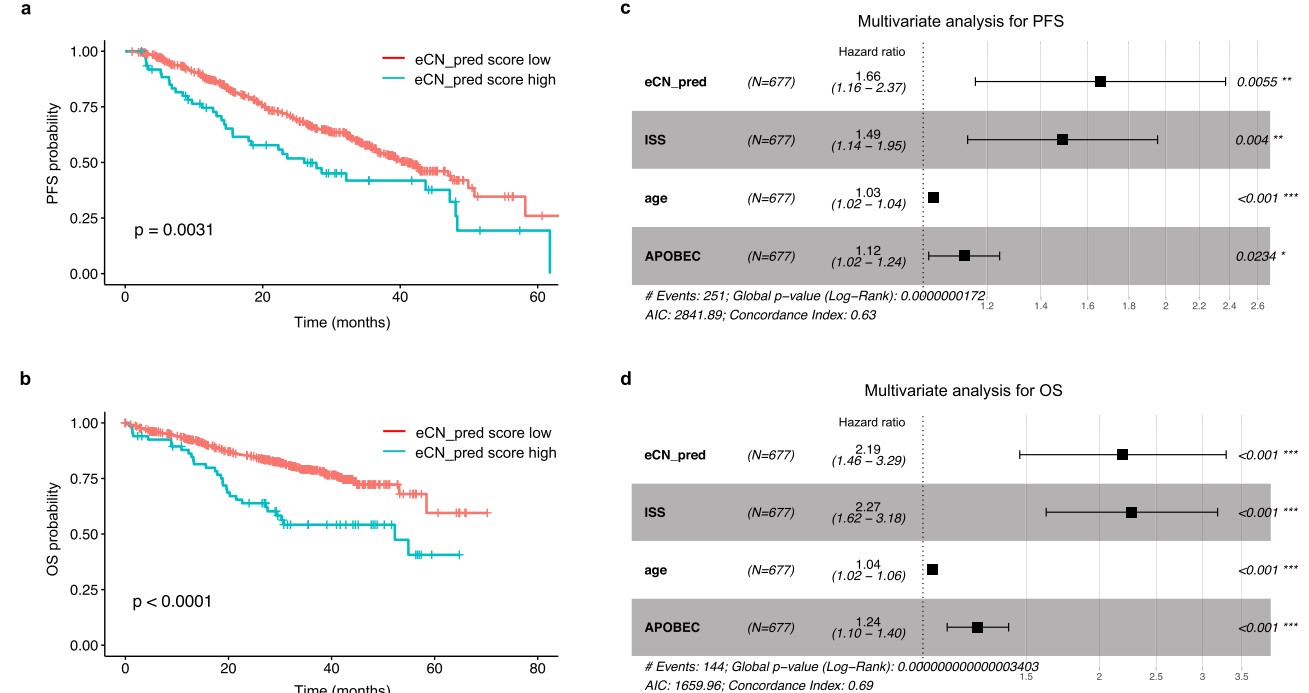

**Fig. 8 Copy number (CN) signatures extracted from whole-exome sequencing (WES) in newly diagnosed multiple myeloma are highly predictive of clinical outcomes. a** Progression-free survival (PFS) probability in the CoMMpass dataset according to high (blue) or low (red) exome CN-prediction score (eCN_pred) for chromothripsis. **b** Overall survival (OS) probability in the CoMMpass dataset according to high (blue) or low (red) eCN_pred. **c** Multivariate analysis of the effect of eCN_pred on PFS after correction for International Staging Score (ISS), age and APOBEC mutational activity. **d** Multivariate analysis of the effect of eCN_pred on OS after correction for the same factors. All p-values for Kaplan–Meier curves were generated according to a 2-sided log-rank test. Multivariate analysis was performed by the Cox proportional hazards model with p-values according to a 2-sided Wald test. Data are presented as median values ± 95% confidence interval.

data, multivariate analysis revealed a significant association between CN signatures and shorter PFS and OS. Indeed, the clinical impact of CN signatures was similar in WGS and WES data. This is important from a translational perspective because it provides an easier pathway towards clinical application in the standard of care setting of NDMM patients. Additional validation would be required to demonstrate in other cancer types the applicability of CN signatures to detect chromothripsis in WES data.

In conclusion, CN signature analysis can accelerate our ongoing quest to accurately define high-risk MM, and to translate WGS-based prognostication into the clinic.

## Methods

**Samples**. All the raw data used in this study are publicly available. Somatic CN profiles for the definition of CN signatures in MM were generated from 752 NDMM patients with low-coverage long-insert WGS (median 4–8×) from the CoMMpass study. The CoMMpass study is a prospective observational clinical trial (NCT01454297) with comprehensive genomic and transcriptomic characterization of NDMM patients, funded and managed by the Multiple Myeloma Research Foundation (MMRF)[40]. The study is ongoing, with data released regularly for research use via the MMRF research gateway. In this study, we used Interim Analysis (IA) 13.

The validation dataset of hematological cancer WGS was compiled from several sources. Data from the PCAWG study[5,29,30] was accessed via the data portal http://dcc.icgc.org/pcawg/, comprising 92 chronic lymphocytic leukemia, 29 chronic myeloid leukemia, 104 B-cell lymphoma and 7 acute myeloid leukemia. An additional 3 therapy-related AML were included, with the WGS data available from European Genome-phenome (EGA) under the accession code EGAD00001005028. Together, these samples formed the non-MM validation WGS set (n = 235, Supplementary Table 6). The MM validation dataset (n = 34) comprised 28 NDMM, 4 monoclonal gammopathy of undetermined significance (MGUS), 2 smoldering MM (SMM) and 1 plasma cell leukemia (PCL). It was compiled from 3 studies that can be accessed from EGA and the database of Genotypes and Phenotypes (dbGAP) with accession codes EGAD00001003309, EGAS00001004467 and phs000348.v2.p1[14,29,41].

WES from 677 NDMM patients were accessed from the CoMMpass study as above, with each patient having concurrent WGS available for comparison. As previously published data, ethics committees or institutional review boards at each of the CoMMpass study sites approved the original study, which was conducted in accordance with the Declaration of Helsinki. All patients provided written informed consent.

**CNV signature analysis**. Genome-wide somatic copy number (CN) profiles were generated from 752 NDMM patients with long-insert low-coverage WGS available from the CoMMpass study. Paired-end reads were aligned to the human reference genome (HRCh37) using the Burrows Wheeler Aligner, BWA (v0.7.8). CN variation and loss-of-heterozygosity events were identified using tCoNuT, a TGen developed tool, with MMRF CoMMpass specific optimizations (https://github.com/tgen/tCoNuT)[9,20], with verification performing using controlFREEC (standard settings for GC content, read size, segmentation, and windows)[21], demonstrating >90% agreement between the two approaches[9]. We minimized the inclusion of artefacts by removing all CN changes smaller than 50kB and excluding the regions corresponding to IgH, IgL, and IgK, as well as the X chromosome from analysis.

The optimal number of categories in each of the 6 CN features detailed in Fig. 1 were established using a mixed effect model with the *mclust* R package, producing a CN category matrix with defined limits for each feature (Supplementary Table 2). Given the lower complexity of MM CN changes compared to the original CN signature definition in ovarian cancer[18], fixed criteria for copy number status were introduced (#1 = bi-allelic deletion; #2 = monoallelic deletion; #3 = diploid; #4 = single gain; 5# = two or more gains, Supplementary Data 1, Supplementary Table 2). De novo CN signature extraction was performed from this matrix via the hierarchical Dirichlet process (*hdp*, https://github.com/nicolaroberts/hdp). The extracted CN signatures were then correlated with publicly available clinical data and manually curated SV data as detailed below to allow the calculation of prediction metrics. The accuracy of chromothripsis prediction from CN signatures was assessed by the area-under-the-curve (AUC) from receiver operating characteristic (ROC) curves via 10-fold cross validation, using all extracted CN signatures as input. The sensitivity and specificity of chromothripsis prediction from varying levels of probability (i.e., AUC) were compared, (Supplementary Table 5), with a prediction level ≥0.6 defining a high CN_pred (WGS) and eCN_pred (WES) score. This score provided the highest level of sensitivity for chromothripsis prediction while still keeping the specificity level at / above 95% for both WGS- and WES-based prediction.

Somatic variant calling was performed using DELLY (v0.7.6)[22] and Manta (v.1.5.0)[23]. The final catalog of high-confidence SVs was obtained by integrating DELLY and Manta calls with copy number data and applying a series of quality filters[9]. Briefly, all SVs called and passed by both callers were included and SVs called by a single caller were only included in specific circumstances: (i) SVs supporting copy-number junctions, (ii) reciprocal translocations, and (iii) translocations involving an immunoglobulin locus (i.e., *IGH, IGK,* or *IGL*).

Single and complex SV events were defined according to the most recent criteria[1–5]. Chromothripsis was defined by more than 10 interconnected SV breakpoint pairs associated with oscillating CN across one or more chromosomes; definition included (i) clustering of breakpoints, (ii) randomness of DNA fragment joins, and (iii) randomness of DNA fragment order across one or more chromosomes. Chromoplexy was defined by interconnected SV breakpoints across >2 chromosomes associated with CN loss. Templated insertions were defined as translocations associated with focal CN gain; if >2 chromosomes were involved templated insertions were classified as complex. Patterns of 3 or more interconnected breakpoint pairs that did not fall into the above categories were classified as "complex", not otherwise specified[9].

The majority of the clinical association data was obtained directly from the CoMMpass data portal (https://research.themmrf.org). The definition of high APOBEC activity was obtained from single-base substitution (SBS) signature analysis; a mutational signature fitting approach using the R package *mmsig*, (https://github.com/evenrus/mmsig) was applied to single nucleotide variant calls from WES data[29,30,42]. High APOBEC mutational activity was defined by an absolute contribution of APOBEC-associated signatures (SBS2 and SBS13) in the top decile, among patients with evidence of APOBEC activity[9,30].

CN variation data from the validation dataset of hematological cancers was utilized for de novo CN signature extraction (hCN-SIG, Supplementary Table 7) without reference to the CoMMpass WGS-derived CN signatures. Fixed criteria for copy number status were introduced as detailed above. The presence of chromothripsis was confirmed for every event by manual inspection and curation of SV and CN data. The accuracy of chromothripsis prediction from was assessed by AUC from ROC curves using all extracted hCN signatures as input. 5-fold cross validation was used for the non-MM cohort prediction, which was then used as the training model for testing the prediction from the MM validation cohort.

CN variation and loss-of-heterozygosity events from the CoMMpass WES sequencing data was assessed using FACETS (Fraction and Allele specific Copy number Estimate from Tumor/normal Sequencing, https://github.com/mskcc/facets)[43]. Fixed criteria for copy number status were introduced as detailed above, then de novo CN signature extraction, and clinical / genomic correlation were all performed without reference to the WGS-derived CN signatures.

**SV signature analysis**. WGS SV data were assessed for clustering, then annotated by SV type and size as per previously described standard criteria (SV type; deletion, tandem duplication, inversion, translocation and size; 1–10 kb, 10–100 kb, 100–1000 kb, 1–10 Mb, >10 Mb)[19]. The resulting matrix of 32 SV features was used for de novo signature extraction by *hdp*, producing 10 SV signatures (Supplementary Data 2, Supplementary Table 4). The accuracy of chromothripsis prediction from the combination of SV and CN signatures, and from SV signatures alone was assessed by AUC from ROC curves via 10-fold cross validation, using all extracted signatures as input (Supplementary Data 3).

**Comparison of CN signatures with alternate CN assessment approaches**. The LOH_index and the GSS were calculated from allele-specific CN files, with the methods being applicable to either WGS or WES data. The LOH_index was calculated using the R package *signature.tools.lib*[38,44] (https://github.com/Nik-Zainal-Group/signature.tools.lib), while the GSS was calculated using the R package *scarHRD*[33] (https://github.com/sztup/scarHRD). The *scarHRD* output is 3 separate CN features (loss-of-heterozygosity, telomeric allelic imbalance, and number of large-scale transitions) which are summed to produce a final score.

To compare chromothripsis-prediction from CN signatures with each of the LOH_index and the GSS, we first calculated with difference in average AUC between two methods estimated from 10-fold cross-validation. Then, standard deviation of the difference in AUCs was estimated by performing a bootstrap resampling. On each new bootstrap sample, we estimated difference in the average AUC between two methods using 10-fold cross-validation. This procedure was repeated 1000 times (Supplementary Data 4).

**Data analysis and statistics**. Analysis was carried out in R version 3.6.1. Key software tools noted throughout the workflow (including *mclust, hdp, survminer, pROC, mmsig, signature.tools.lib,* and *scarHRD*) are publicly available. Unless otherwise specified, we used the Wilcoxon rank-sum test to test for differences in continuous variables between two groups and Fisher's exact test for 2 × 2 tables of categorical variables. Associations between clinical, molecular characteristics, and survival were evaluated by fitting Cox proportional hazard models, with the p-values reported based on the Wald test.

**Reporting summary**. Further information on research design is available in the Nature Research Reporting Summary linked to this article.

## Data availability

All sequencing BAM files are available at the EGA and dbGaP archives under accession codes as listed below. The CoMMpass dataset, while publicly available, requires that access is requested via https://research.themmrf.org/ rather than raw data being published directly. All other data are available under restricted access, with access obtained by contacting the public depository listed. WES and low coverage/long insert WGS sequencing data from 752 NDMM patients (CoMMpass trial; IA 13) under phs000748.v1.p1. WGS from 24 NDMM and 1 high risk SMM patient under EGAD00001003309 and phs000348.v2.p1. WGS data from 3 MM, 1 SMM, 1 PCL, and 4 MGUS patients under EGAS00001004467. WGS data from 3 therapy related AML patients under EGAD00001005028. Data from the PCAWG study was accessed via the data portal: https://dcc.icgc.org/. Source data are provided with this paper.

## Code availability

The analytical workflow in R for the de novo extraction of CN signatures is provided in Supplementary Data 1 and of SV signatures is in Supplementary Data 2. The code for predicting chromothripsis from genomic signatures is detailed in Supplementary Data 3 and the approach to comparing 2 methods for predicting chromothripsis is presented in Supplementary Data 4. All code is also available on Zenodo (https://zenodo.org/record/5095351#.YO2szi9h04c, https://doi.org/10.5281/zenodo.5095351).

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

## Acknowledgements

This work is supported by the Multiple Myeloma Research Foundation (MMRF), the Perelman Family Foundation, the Riney Family Multiple Myeloma Research Program Fund, a Memorial Sloan Kettering Cancer Center NCI Core Grant (P30 CA 008748), and by a Sylvester Comprehensive Cancer Center NCI Core Grant (P30 CA 240139). F.M. is supported by the American Society of Hematology, the International Myeloma Foundation, and The Society of Memorial Sloan Kettering Cancer Center. K.H.M. is supported by the Haematology Society of Australia and New Zealand, the Royal College of Pathologists of Australasia, the Royal Australasian College of Physicians, the Snowdome Foundation, and the Multiple Myeloma Research Foundation. G.J.M. is supported by The Leukemia Lymphoma Society. N.B. is supported by the European Research Council under the European Union's Horizon 2020 research and innovation programme (grant agreement No. 817997)

## Author contributions

F.M. designed and supervised the study, collected and analyzed data and wrote the paper; O.L. supervised the study, collected and analyzed data and wrote the paper; K.H.M. collected, analyzed and interpreted the data and wrote the paper; G.J.M. interpreted the data and wrote the paper; E.H.R., A.De., Z.B., V.Y. and B.D. collected and analyzed data. M.H., B.Z., E.M.B., P.B., N.B., Y.Z., A.Do. and A.M.L. collected data.

## Competing interests
