## [Peer Review File · Nature Communications]

REVIEWER COMMENTS

Reviewer #1 (Remarks to the Author): Expert in myeloma genomics

In this paper, Dr. Maclachlan and colleagues sought to predict the presence of chromothripsis in multiple myeloma (MM) by applying copy number (CN) signatures. They used the CoMMpass dataset to analyze 752 WGS samples as well as a validation cohort of 677 WES samples. They mainly observed 5 CN-signatures in MM, among which the #4 and 5 correlate with MAF/MAFB translocations, APOBEC signatures, TP53 inactivation, gain of 1q and the presence of chromoplexy and chromothripsis. They further defined a CN-signature predictive model and validated its ability to predict the presence of chromothripsis as well as its clinical impact on PFS and OS. This was also applicable when using a WES cohort to define CN-signatures.

The manuscript is well written and the data are clearly exposed.

The work is of great interest as genomic instability and complex structural variants represent a major driver of progression in MM. Tools to infer the presence of chromothripsis in clinical practice would be therefore very informative. In that sense, the present work has a clear clinical application even though WES is not yet widely used in routine practice. In a more functional aspect, I believe that the manuscript would gain interest with more focus on the underlying biological insight: how these CN-signatures and their correlation to complex structural variants may provide new knowledge on MM oncogenesis?

Regarding the CN-signatures, do you have any indication of their phylogeny? Are there key CN features associated with early vs late events? Are they clonal or subclonal? What is their correlation with mutational signatures? Do you observe MM drivers more often associated with specific CN-signatures?

Similarly, by analyzing the DNA sequence at breakpoints, do you see specific patterns of repair process that characterize one signature vs another?

Regarding the clinical applicability of the work - how did you define the CN-pred score? Is a high CN-pred score driven only by the presence of CN-sig 4 or 5? Can it be applied in clinical practice to a single patient? For instance, in the example Figure 4a and b, I believe the pattern correspond to a CN-sig 5, which likely represent a high CN-pred score? Do you also observe intermediate patterns and how do you classify them?

Finally, since you have two important cohorts of MM patients, among the ones with chromothripsis (or a high CN-pred score) do you see an impact of the type of treatment they received on their outcome (ASCT vs no, daratumumab vs no, PI vs no, IMiDs vs no...)?

Reviewer #2 (Remarks to the Author): Expert in bioinformatics, computational genomics, and chromothripsis

Maclachlan et al present an interesting analysis of chromothripsis and copy number patterns in newly diagnosed multiple myeloma patients using data from the CoMMpass trial. The authors built upon their previous work and the recently proposed analytical framework of 'copy number signatures' to identify patients with chromothripsis using either WGS or WES data. Although the

study represents a nice effort to link genome sequencing data with clinical outcome variables, it is far from mature for publication. My main concerns are the following.

Major:

- The tests used throughout the paper are not indicated. Details about the statistical tests used should accompany all P values reported. For instance, in section "CN signatures are strongly predictive of chromothripsis in multiple myeloma" the test used to compute the P values should be indicated. In other cases, the statistical analysis underpinning the conclusions is not described. For instance, in: "Patients with chromothripsis events were characterized by poor clinical outcomes, with chromothripsis being associated with multiple unfavorable" The statistical significance for these claims/associations is missing. How did the authors test that association?
- Do the results (significant associations) reported in figure 3 hold if the analysis is done for sigs 4 and 5 separately, rather than together? From figure 3a it seems that the association between signature 5 and chromothripsis might be weak. This is critical as some of the features associated with signature 5 would be consistent with repair-associated complex rearrangements rather than chromothripsis arising from random rejoining of DNA fragments after fragmentation.
- Why predict chromothripsis necessarily, rather than patterns of genome instability or complex events? Results/data showing that chromothripsis specifically, rather than genome instability, is more predictive of prognosis should be clearly reported.
- The copy number signature analysis builds upon previous work by the Brenton group. Have the authors considered the inclusion of minor copy number information to derive the signatures? Such information could be critical to distinguish real chromothripsis events from clustered rearrangements (e.g. clusters of deletions or duplications). I acknowledge that the authors use the LOH_index for comparison – I am referring to the inclusion of minor copy number information for signature derivation.
- A substantial number of cases have both 'complex patterns' and 'chromothripsis' (Figure 3). Thus, I wonder whether the associations between chromothripsis and patient prognosis that the authors are discovering are related to genome instability, irrespective of the mechanism generating it, rather than to chromothripsis specifically.
- Regarding the performance of the signature-based algorithm for chromothripsis detection, what is the baseline method for comparison? How much better do the signatures do against sample stratification based on e.g. the number of SVs per sample, or other proxy variables for genome instability? These comparisons should be included as a baseline.
- Regarding the validation of the signature-based detection of chromothripsis, a discussion of the cases missed by the predictive algorithm is missing. Which type of events are not picked up by the signature-based algorithm? Which ones are over/miscalced?
This is also related to a major concern: what are the added advantages of calling chromothripsis using the signature-based method for WGS data? Why is this approach better than others (ChromAL, ShatterSeek, etc..) and in which cases? This is not evident from the study at present. If the authors are presenting a new chromothripsis detection method, it should be benchmarked against state-of-the-art algorithms developed for this task. Answering that the approach also works for WES data is

not satisfactory (see also below) – the novelty of such approach would be questionable and the validation performed so far would not guarantee extrapolation to other cohorts.

- In the validation exercise performed by the authors (section “CN signatures are strongly predictive of chromothripsis in hematological malignancies”), how many cases had chromothripsis? PCAWG reported chromothripsis and copy number calls for all cases. A validation using these data should be included, in particular because the number and diversity of chromothripsis calls for hematological malignancies was very low. Importantly, the authors should carefully examine whether their method is capable of discriminating classical (a.k.a. canonical) chromothripsis events from oscillating copy number patterns generated by homologous recombination deficiency. This has been a major challenge for chromothripsis calling in the field. Although the focus of this study is on MM, the generalization capability of the method and framework should be evaluated using other data sets, which is a low-effort exercise given that, as said above, calls are readily available.

- Similarly, there is a missed opportunity here to validate the WES-based calling of chromothripsis using data from TCGA. About 800 samples from TCGA have both WES and WGS data, so a validation using those cases should be presented (see Bailey et al Nat Comm 2020). The copy number calls for both WES and WGS data sets have been published by the Pan-Cancer Atlas and PCAWG projects. Developing a new method just focused on MM would not be novel enough for publication in Nat Comm.

- In figure 3 there are multiple events per sample. In addition to the covariates used in the Cox models, how are these other variables (e.g. ploidy) affecting the results? Were these clinical variables considered in the ‘feature selection’ step?

- In Figure 4a, what are the “6” and “9” in the plot?

- Regarding the analysis reported in Figure 5, how do these results compare to patient stratification based on other variables related to genome instability, such as the presence of complex rearrangements, percentage of the genome altered, number of SVs, etc? Are the results still significant after including other covariates accounting for genome instability in the model? That is, does the signal from chromothripsis stand out even if ploidy, percentage of genome altered, etc.. are included in the model?

- Chromothripsis is treated as a binary variable in this study. Although the chromothripsis patterns might be less complex than those observed in other malignancies, e.g. sarcomas, chromothripsis patterns have been classified in distinct categories, and pan-cancer analysis of this mechanism have certainly revealed a very complex and varied landscape. Could the authors elaborate more on the types of chromothripsis detected in their cases, further stratify based on timing (e.g. before and after WGD), etc..? Referring to a previous publication is not acceptable given the central role of chromothripsis in this study. Treating chromothripsis as a binary variable is an oversimplification of this mechanism and its consequences.

- From Figure 3a one could argue that the association between signature exposure might also hold for chromoplexy and/or complex rearrangements. Is this the case? How much predictive power do sigs 4 and 5 have for chromothripsis as compared to complex events/chromoplexy?

- A major issue of this study is that the description of the methods used is insufficient. For instance:

What algorithm was used to call somatic copy number aberrations using the WGS data? How were the copy number and ploidy calls validated? What parameter were chosen to run controlFREEC and how was the 'verification' performed? How was the ploidy set up in controlFREEC? Etc..

The methods section related to the selection of the optimal number of categories in each of the 6 CN features using a mixed effect model should be extended to allow future readers to follow/reproduce all steps of the analysis.

The methodology for chromothripsis detection, which is key to his study, is not described in sufficient detail. This should be addressed carefully. The software and parameters used for statistical significance are not included. Details on the validation for the cut-off values used are also missing. Etc. How do their calls compare to other chromothripsis detection methods such as ChromAL (Notta et al 2016) or ShatterSeek (Cortes-Ciriano, Nat Genet 2020). Even if the authors are using data from a previous study this information should be included here.

Similarly, the detection of chromoplexy is defined as: "Chromoplexy was defined by interconnected SV breakpoints across >2 chromosomes associated with CN loss."

What algorithm was used? How were these cut-off values validated? While chromoplexy can be associated to copy number loss at the breakpoints forming the chain this is not always the case; how was this handled? How do these results compare to standard chromoplexy detection methods, such as ChainFinder (Baca et al 2013)?

Minor:

In the Introduction: "Although the biological impact is likely similar across various malignancies,". This claim is not substantiated by any data. A reference is needed or otherwise I suggest removing.

Re: Copy number signatures predict chromothripsis and associate with poor clinical outcomes in patients with newly diagnosed multiple myeloma

Thank you for considering our manuscript.

Below, we include a point-by-point response to the reviewer comments – the reviewer comments are in black, our response in blue.

=====

REVIEWER COMMENTS:

Reviewer #1 (Remarks to the Author): Expert in myeloma genomics

In this paper, Dr. Maclachlan and colleagues sought to predict the presence of chromothripsis in multiple myeloma (MM) by applying copy number (CN) signatures. They used the CoMMpass dataset to analyze 752 WGS samples as well as a validation cohort of 677 WES samples. They mainly observed 5 CN-signatures in MM, among which the #4 and 5 correlate with MAF/MAFB translocations, APOBEC signatures, TP53 inactivation, gain of 1q and the presence of chromoplexy and chromothripsis. They further defined a CN-signature predictive model and validated its ability to predict the presence of chromothripsis as well as its clinical impact on PFS and OS. This was also applicable when using a WES cohort to define CN-signatures.

The manuscript is well written and the data are clearly exposed.

The work is of great interest as genomic instability and complex structural variants represent a major driver of progression in MM. Tools to infer the presence of chromothripsis in clinical practice would be therefore very informative. In that sense, the present work has a clear clinical application even though WES is not yet widely used in routine practice. In a more functional aspect, I believe that the manuscript would gain interest with more focus on the underlying biological insight: how these CN-signatures and their correlation to complex structural variants may provide new knowledge on MM oncogenesis?

We thank the Reviewer for their comments. Our previous manuscript (Rustad et al. Blood Cancer Discovery 2020, PMID: 33392515) established the biological and clinical relevance of complex structural variants and chromothripsis in MM; demonstrating that chromothripsis is independently associated with shorter progression-free and overall survival. Furthermore, few weeks ago we provided more details regarding the role of structural variants and complex events such as chromothripsis in the progression of myeloma precursor conditions (Oben et al. Nat Comm 2021; PMID: 33767199).

In this study, the primary aim was to demonstrate how CN signatures are able to predict chromothripsis, both in whole genome (WGS) and whole exome sequencing (WES) without requiring structural variation data. From our previous studies we know that

chromothripsis is associated with many unfavorable MM genomic features such as: APOBEC mutational activity, del17p and *TP53* mutations, 1q gain and 1q amp. In the current study we investigated all these different biological associations in order to demonstrate that CN signatures, in particular CN-SIG4 and CN-SIG5, are faithful to the adverse prognostic features associated with chromothripsis. We have expanded the background presentation of the clinical relevance of chromothripsis in the updated manuscript (pages 3-4 lines 67-81).

- Regarding the CN-signatures, do you have any indication of their phylogeny? Are there key CN features associated with early vs late events? Are they clonal or subclonal?

The Reviewer pointed out a very important and interesting aspect. Unfortunately, the MMRF samples were sequenced using a long-insert low-coverage WGS platform (median coverage 4-8x) which does not provide sufficient depth to define subclonal events and to accurately reconstruct the phylogenetic tree of each case. However, the early onset and the clonal nature of chromothripsis in MM is supported by several lines of evidence:

- 1) Chromothripsis can be detected in precursor conditions (monoclonal gammopathy of uncertain significance, MGUS and smoldering myeloma, SMM) years before progression to MM, representing an early genomic MM-defining event which is highly predictive for later progressive disease (Oben et al. Nat Comm 2021; PMID: 33767199 and Bolli et al. Nat Comm 2018, PMID: 30135448).
- 2) The molecular time of the large gains caused by chromothripsis are very early, usually acquired 2-4 decades before MM diagnosis (Maura et al. Nat Comm 2019, PMID: 31444325)
- 3) Using longitudinal full coverage WGS data, we have previously shown how chromothripsis remains relatively stable over time during MM evolution from precursor condition, to MM diagnosis, to relapse, and death (Maura et al. Nat Comm 2019, PMID: 31444325 and Landau et al. Nat Comm 2020, PMID: 32680998).
- 4) In this study we also showed how the chromothripsis and its related CN features are relatively stable over time in paired samples collected before after treatment within the MMRF WGS trial.

These data and considerations are now included in the updated version of the manuscript (pages 3-4; lines 67-81, and new Supplementary Figure 1).

- What is their correlation with mutational signatures? Do you observe MM drivers more often associated with specific CN-signatures?

CN-SIG4 and CN-SIG 5 (containing CN features consistent with chromothripsis) were associated with several high-risk myeloma genomic defining events. In regard to the mutational signatures, we observed a significant association between these CN-SIG4

and CN-SIG 5 and APOBEC mutational activity (single base substitution signatures 2 and 13, Figure 3). APOBEC is known to be associated with poor prognosis, *TP53* mutations and unfavorable translocations (*MAF-* and *MAFB-IGH*). The association between CN-SIG4 and CN-SIG 5 and APOBEC further support the ability of CN signatures to predict presence of chromothripsis.

- Similarly, by analyzing the DNA sequence at breakpoints, do you see specific patterns of repair process that characterize one signature vs another?

This analysis would require that all CN breaks are supported by a structural variant able to define the genomic position and motif. In fact, the CN coordinates usually represent the last germline SNP before the breakpoint. While in the CoMMpass dataset SVs justify 83% of all intrachromosomal CNAs, several whole chromosome and CN changes are not justified, preventing a comprehensive investigation.

Regarding a more focused SV breakpoint investigation, we did not observe any clear association between chromothripsis and either non-homologous end joining, alternative end joining or microhomology-mediated break-induced replication.

- Regarding the clinical applicability of the work - how did you define the CN-pred score? Is a high CN-pred score driven only by the presence of CN-sig 4 or 5? Can it be applied in clinical practice to a single patient? For instance, in the example Figure 4a and b, I believe the pattern correspond to a CN-sig 5, which likely represent a high CN-pred score? Do you also observe intermediate patterns and how do you classify them?

The prediction score was defined from the area-under-the-curve (AUC) from receiver operating characteristic (ROC) curves via 10-fold cross validation, using all extracted CN signatures as input, with a probability ≥ 0.6 defining a high CN_pred (WGS) and eCN_pred (WES) score. The level of 0.6 come from testing the sensitivity and specificity from different levels of probability. As expected, the prediction is largely driven by CN-SIG4 and CN-SIG5. In the revised version of the manuscript we specify these aspects in the Methods section (page 20; lines 448-455). We also include the sensitivity and specificity of chromothripsis prediction from varying levels of probability in Supplementary Table 5.

Regarding the question if it is possible to use this approach on one single patient, the answer is yes, assuming a cohort was available with a similar sequencing methods and depth.

Regarding intermediate patterns and discrepancy between manual inspection and CN-signature prediction, in the new version of the manuscript, we demonstrate that the CN signature-based prediction is most accurate in samples containing higher complexity chromothripsis, with lower complexity events being more likely to be mis-classified. However, the adverse survival associated with chromothripsis was mostly confined to true positive cases, indicating that CN signatures capture the most biologically relevant event. These analysis and data have been included in the new version of the

manuscript in Supplementary Figures 6 and 11, and at page 10; lines 224-234 and page 13; lines 283-290.

- Finally, since you have two important cohorts of MM patients, among the ones with chromothripsis (or a high CN-pred score) do you see an impact of the type of treatment they received on their outcome (ASCT vs no, daratumumab vs no, PI vs no, IMiDs vs no...)?

In our previous study, (Rustad et al. Blood Cancer Discovery 2020; PMID: 33392515), the poor prognostication of chromothripsis was independent of all clinical, therapeutical and biological features. In line with that, CN prediction retained its significance even when corrected for key therapy exposure (see new Supplementary Figure 10).

Reviewer #2 (Remarks to the Author): Expert in bioinformatics, computational genomics, and chromothripsis

Maclachlan et al present an interesting analysis of chromothripsis and copy number patterns in newly diagnosed multiple myeloma patients using data from the CoMMpass trial. The authors built upon their previous work and the recently proposed analytical framework of 'copy number signatures' to identify patients with chromothripsis using either WGS or WES data. Although the study represents a nice effort to link genome sequencing data with clinical outcome variables, it is far from mature for publication. My main concerns are the following.

Major:

- The tests used throughout the paper are not indicated. Details about the statistical tests used should accompany all P values reported. For instance, in section "CN signatures are strongly predictive of chromothripsis in multiple myeloma" the test used to compute the P values should be indicated. In other cases, the statistical analysis underpinning the conclusions is not described.

We agree with the Reviewer that details regarding statistical tests and p-values should have been provided in the figures. In the previous version we specified the statistical test used in the Method section: "*Unless otherwise specified, we used the Wilcoxon rank sum test to test for differences in continuous variables between two groups and Fisher's exact test for 2x2 tables of categorical variables*". Following Reviewer comments, we added this relevant information in each figure legend.

- For instance, in: "Patients with chromothripsis events were characterized by poor clinical outcomes, with chromothripsis being associated with multiple unfavorable" The statistical significance for these claims/associations is missing. How did the authors test that association?

This statement, indicating data established in our previous publication, (Rustad et al. Blood Cancer Discovery 2020; PMID: 33392515), has been moved to the introduction.

- Do the results (significant associations) reported in figure 3 hold if the analysis is done for sigs 4 and 5 separately, rather than together? From figure 3a it seems that the association between signature 5 and chromothripsis might be weak. This is critical as some of the features associated with signature 5 would be consistent with repair-associated complex rearrangements rather than chromothripsis arising from random rejoining of DNA fragments after fragmentation.

The associations in Figure 3 hold true for CN-SIG4 alone, but CN-SIG5 is too sparse to examine as an independent factor. However, we do note that the addition of CN-SIG5 adds to the prediction of chromothripsis, with the AUC for CN-SIG4 alone being 0.88 (AUC difference=0.03, standard deviation=0.01, p=0.009, based on bootstrap analysis). This has now been specified in the updated version of the manuscript (page 9; lines 208-12).

- Why predict chromothripsis necessarily, rather than patterns of genome instability or complex events? Results/data showing that chromothripsis specifically, rather than genome instability, is more predictive of prognosis should be clearly reported.

The Reviewer pointed out a very important aspect of MM biology. In contrast to the patterns reported in solid cancers (e.g. Isidro Cortés-Ciriano et al. Nat Gen 2019, PMID: 32025003), in MM, chromothripsis is usually acquired at very early stage of cancer development, maintaining its structure as relatively stable over time.

This model is supported by multiple lines of evidence:

- 1) Chromothripsis can be detected in MM precursor conditions (monoclonal gammopathy of uncertain significance, MGUS and smoldering myeloma, SMM) years before progression to MM, representing an early genomic MM-defining event which is highly predictive for later progressive disease (Oben et al. Nat Comm 2021; PMID: 33767199 and Bolli et al. Nat Comm 2018, PMID: 30135448).
- 2) The molecular time of the large gains caused by chromothripsis are very early, usually acquired 2-4 decades before MM diagnosis (Maura et al. Nat Comm 2019, PMID: 31444325 and Rustad et al. Nat Comm 2020, PMID: 32317634).
- 3) Interestingly, some of the chromothripsis events are linked to and responsible of IGH-translocations known to be key MM initiating events (Rustad et al. Blood Cancer Discovery 2020, 33392515)
- 4) Using longitudinal full coverage WGS data, we have previously showed how chromothripsis remains relatively stable over time during the evolution from precursor condition, to MM diagnosis, to relapse, and death (Maura et al. Nat Comm 2019, PMID: 31444325 and Landau et al. Nat Comm 2020, PMID: 32680998).

- 5) In this study we also showed how the chromothripsis and its related CN features relatively stability over time in paired samples collected before after treatment is observed in the MMRF WGS (new Supplementary Figure 1).

Similarly to other hematological cancers, MM genome is less complex compared with solid cancers. In the majority of MM, patterns of chromosomal instability such as BRCA, MSI, whole genome duplication, double minutes, bridge-break-fusion, typhonas, rigma and pyros are absent or extremely rare.

In MM, we often observe significant genomic complexity in patients with high APOBEC activity, *TP53* bi-allelic inactivation and amp1q. While chromothripsis is associated with each of these features, many patients with chromothripsis (91/178, 51%) don't harbor any of them, meaning that chromothripsis doesn't reflect just these processes. Furthermore, APOBEC, amp 1q, and *TP53* mutations are known to be late events in MM, often subclonal and selected after treatment (Maura et al. Nat Comm 2019, PMID: 31444325; Rustad et al. Nat Comm 2020, PMID: 32317634; Walker et al. Blood 2018, PMID: 29884741; and Bolli et al. Nat Comm 2014, PMID: 24429703). In this model, chromothripsis proceeds the onset of these additional high-risk features, suggesting that this catastrophic event is independent from later genomic complexity / instability.

Data regarding these relevant aspects of MM biology are now included in the updated version of the paper (pages 3-4; lines 67-81, and new Supplementary Figure 1).

- A substantial number of cases have both 'complex patterns' and 'chromothripsis' (Figure 3). Thus, I wonder whether the associations between chromothripsis and patient prognosis that the authors are discovering are related to genome instability, irrespective of the mechanism generating it, rather than to chromothripsis specifically.

Regarding the co-occurrence of complex SV (not otherwise specified; NOS) and chromothripsis, and the association between CN-signatures and genomic complexity/instability, following Reviewer suggestions we demonstrate how:

- 1) Similarly to *ShatterSeek*, CN-signatures are able to accurately predict the more complex chromothripsis events, but lose sensitivity for the less complex
- 2) In patients with chromothripsis, the CN- and SV-signature landscape of non-chromothripsis events did not show any enrichment for features comprising CN-SIG4 and CN-SIG5, suggesting that additional complex or single events do not increase the chromothripsis-related signatures. This confirms the idea that in hematological malignancies the genomic background on which chromothripsis occurred is simpler compared with solid cancers. To address these aspects, we included a new Supplementary Figure showing how the non-chromothripsis SV and CN landscape of patients with chromothripsis is not significantly different from that observed in patients without chromothripsis (new Supplementary Figure 7; cosine similarity 0.98 for CN profile and 0.96 for SV profile).
- 3) The association between complex-NOS and CN-signatures is partially driven by the false negative events. In fact, similarly to Maciejowski et. al. Nat Gen 2020,

PMID: 32719516, we used 10 SV as threshold for chromothripsis. It is possible that some events annotated by complex are actually chromothripsis events, or as called by Maciejowski et. al., “chromothripsis-like events”. These events having less than 10 SV, will affect the CNV profile less, making detection more difficult.

Overall, the idea that chromothripsis occurs within complex and unstable myeloma genomes is not supported by our data and by the previous literature. This concept is now included in the updated version of the manuscript (page 10; lines 213-223).

- From Figure 3a one could argue that the association between signature exposure might also hold for chromoplexy and/or complex rearrangements. Is this the case? How much predictive power do sigs 4 and 5 have for chromothripsis as compared to complex events/chromoplexy?

While CN-SIG4 and CN-SIG5 are associated with both complex-NOS and chromoplexy using a Wilcoxon test, their prediction is low (each AUC=0.74) compared with chromothripsis (AUC=0.90). (Please see new Supplementary Figure 5).

- The copy number signature analysis builds upon previous work by the Brenton group. Have the authors considered the inclusion of minor copy number information to derive the signatures? Such information could be critical to distinguish real chromothripsis events from clustered rearrangements (e.g. clusters of deletions or duplications). I acknowledge that the authors use the LOH_index for comparison – I am referring to the inclusion of minor copy number information for signature derivation.

In adapting the CN signature approach from that already published, we made only the necessary adaptations to reflect distinct MM biology features (e.g. the immunoglobulin loci and specifying hyperdiploidy), otherwise keeping to the original description as much as possible. The inclusion of the minor allele might improve the approach but would also create something different compared to the previous work from Brenton group. Considering this, we decided to keep the approach as close as possible to the original described.

- Regarding the performance of the signature-based algorithm for chromothripsis detection, what is the baseline method for comparison? How much better do the signatures do against sample stratification based on e.g. the number of SVs per sample, or other proxy variables for genome instability? These comparisons should be included as a baseline.

In MM, the association between chromothripsis and genomic instability is not as strong as in solid cancers. MM usually acquires chromothripsis very early in time, with these complex events being conserved and unchanged during different evolutionary phases. While SV represents one of the key drivers in MM evolution and drug resistance, complex events like templated insertion and chromothripsis are rarely subclonal and selected at relapse (e.g. Supplementary Figure 1 and Maura et al. Nat Comm 2019,

PMID: 31444325). Other events, mostly single SVs, play a major role in defining late MM drivers. Overall, the concept of solid cancer genomic instability doesn't apply to most hematological cancer bulk NGS data. In the new version of the manuscript, we show that the genomic background outside of the chromothripsis events is not enriched for either features of genomic instability nor features that might increase the CN-SIG4 and CN-SIG5 contribution (see Supplementary Figure 7, and text page 10 lines 213-223).

-This is also related to a major concern: what are the added advantages of calling chromothripsis using the signature-based method for WGS data? Why is this approach better than others (ChromAL, ShatterSeek, etc..) and in which cases? This is not evident from the study at present. If the authors are presenting a new chromothripsis detection method, it should be benchmarked against state-of-the-art algorithms developed for this task. Answering that the approach also works for WES data is not satisfactory (see also below) – the novelty of such approach would be questionable and the validation performed so far would not guarantee extrapolation to other cohorts.

The Reviewer raised an important consideration. *ShatterSeek* likely represents the gold standard for chromothripsis prediction in cancer (Isidro Cortés-Ciriano et al. Nat Gen 2019, PMID: 32025003). This approach has been developed based on predominantly solid cancer WGS data and is based on the combination of SV and CNV features. The integration of SV obviously provides an advantage for the prediction of chromothripsis vs CNV alone. To address these relevant points:

- 1) We showed that *ShatterSeek* performed very well in predicting chromothripsis within the CoMMpass data set (AUC=0.93; new Figure 5b). In particular, it efficiently predicted the most complex (AUC=0.95 from cases with >50 chromosomal breakpoints (37/752; 5%). This association also supports the high quality of our SV calls.
- 2) To model a genomic signature approach, similar to what has been done in breast cancer with the BRCAness score by Serena Nik-Zainal (Davies H, et al. Nat Med 2017, PMID: 28288110), we integrated CN and SV signatures predicting chromothripsis with higher accuracy than either SV and CN alone (new Figure 5a).
- 3) The accuracy of the SV-CNV signature approach was analogous to *ShatterSeek*, however, CN signatures alone remained highly accurate (new Figure 5d). The difference in chromothripsis prediction between CN signatures alone and *ShatterSeek* is not statistically significant, based on bootstrap analysis. This suggests that CN signatures can be used in hematological cancers to predict chromothripsis. This is mostly due to the less complex genomic background and absence of “canonical” genomic instability in most of the patients, irrespective of the chromothripsis event (see above).

The advantage of CN signatures in predicting chromothripsis compared with *ShatterSeek* or with CN-SV signatures using WGS data is obviously limited. However, the fact that CN signatures don't need SV, allows their usage on different sequencing

platform (targeted, exomes, single cell RNA). In this paper, having a large cohort of paired WGS and WES, we tested exomes to establish the method, comparing with the ground truth obtained on WGS. Therefore, after successfully comparing *ShatterSeek* and CN-signatures, and working on large cohort of patients, we believe that the ability of this approach to predict chromothripsis from non-WGS is actually an important novelty.

To further prove to the Reviewer how this approach can be effective, we include in this rebuttal an analysis performed using NGS targeted data from 113 newly diagnosed patients, (Yellapantula et al. Blood Cancer J. 2019, PMID: 31827071). Running the same computational workflow, we observed that CN-signatures extracted from targeted have a strong role in defining a high-risk population (see Figure below, with CN-SIG2 in targeted sequencing containing CN features consistent with chromothripsis).

Survival by copy number signatures in multiple myeloma targeted sequencing

Figure legend: (a) Progression-free survival (PFS) according to targeted-sequencing-based copy number signatures (tCN-SIG2, contribution below the median; red, above the median; blue). (a) Multivariate analysis for PFS according to contribution from tCN-SIG2 after correction for age, International Staging Score (ISS), gain or amplification of 1q21, mutation of TP53 or translocation t(4;14) (t_MMSET).

Not having paired WGS data for these patients and therefore not being able to establish the ground truth regarding the presence of chromothripsis, we decided to not include this targeted-sequencing-based CN signature analysis in the final manuscript.

- Regarding the validation of the signature-based detection of chromothripsis, a discussion of the cases missed by the predictive algorithm is missing. Which type of events are not picked up by the signature-based algorithm? Which ones are over/miscalced?

We want to thank the Reviewer for having raised this important point. Following Reviewer's suggestions, we demonstrate that, similarly to *ShatterSeek*, the CN signature-based prediction is most accurate in samples containing higher complexity chromothripsis, with lower complexity events being more likely to be mis-classified. Interestingly, the adverse survival associated with chromothripsis was mostly confined to true positive cases, indicating that CN signatures capture the most biologically and unfavorable relevant event. This analysis and data have been added in the new version of the manuscript (Supplementary Figures 6, 9, and 11, page 10; lines 213-223 and pages 13; lines 283-290).

- In the validation exercise performed by the authors (section “CN signatures are strongly predictive of chromothripsis in hematological malignancies “), how many cases had chromothripsis? PCAWG reported chromothripsis and copy number calls for all cases. A validation using these data should be included, in particular because the number and diversity of chromothripsis calls for hematological malignancies was very low. Importantly, the authors should carefully examine whether their method is capable of discriminating classical (a.k.a. canonical) chromothripsis events from oscillating copy number patterns generated by homologous recombination deficiency. This has been a major challenge for chromothripsis calling in the field. Although the focus of this study is on MM, the generalization capability of the method and framework should be evaluated using other data sets, which is a low-effort exercise given that, as said above, calls are readily available.

We believe that, according to our data and previous published work, the role and profile of chromothripsis in hematological cancers is different compared to solid cancers.

Specifically:

- the number of SV and CNV associated to these events in MM is usually low (see new Supplementary Figure 2).
- The maximum number of focal gains associated with chromothripsis rarely exceed 10 in hematological cancers (see new Supplementary Figure 2).
- Genomic drivers associated with genomic instability (TP53, ATM etc), are often subclonal, suggesting a later onset compared with chromothripsis

In addition, as discussed above, the solid cancer features and hallmarks of genomic instability are absent (or extremely rare) in hematological cancers, and in particular in multiple myeloma. Specifically:

- 1) Combining all hematological WGS included in this study (n=976) only one case of diffuse large B-cell lymphoma had evidence of BRCA1 deficiency, with widespread tandem duplications. Other features of genomic instability have not been reported in hematological cancers.
- 2) In MM, we have previously demonstrated that the homologous recombination deficiency observed in solid cancer is not present (Maura F, et al. Nat. Comm 2019; PMID: 31278357)
- 3) MM doesn't have any of the single-base substitution / doublet base substitution / indel / structural variant signatures associated with patterns of genomic instability in solid cancers (<https://cancer.sanger.ac.uk/signatures/>).

To specifically answer to the Reviewer's questions: in the hematological cancer validation set, 29/235 (12.3%) had chromothripsis, mostly therapy related acute myeloid leukemia and diffuse large B-cell lymphomas.

This study is focused on hematological cancers. Putting aside our lab expertise in these conditions, we believe that the lower hematological cancers complexity represents a key

point. Having a simpler genomic background and absence of genomic instability hallmarks, CN-signatures can provide a particularly accurate information.

Performing preliminary analyses with CN signatures in the whole PCAWG dataset, it is clear that the presence of different patterns of genomic instability, whole genome duplication, and more complex events create serious problems in the definition of the different CN classes. While CN-signatures are accurate at predicting BRCA-deficiency in solid cancers, the prediction of chromothripsis among a highly complex background is less accurate. CN-signature-based chromothripsis prediction works particularly well in hematological cancers due to the simpler genomic background and absence of canonical genomic instability, with chromothripsis being the most complex event that most hematological cancers acquire. To extend our signature-based prediction of chromothripsis to solid cancers would require adaptation for each tumor biological subtype and category and is out of the scope of this paper.

These concepts have been emphasized in the updated version of the manuscript (pages 6-7, lines 137-150; page 10, 213-223).

- Similarly, there is a missed opportunity here to validate the WES-based calling of chromothripsis using data from TCGA. About 800 samples from TCGA have both WES and WGS data, so a validation using those cases should be presented (see Bailey et al Nat Comm 2020). The copy number calls for both WES and WGS data sets have been published by the Pan-Cancer Atlas and PCAWG projects. Developing a new method just focused on MM would not be novel enough for publication in Nat Comm.

We agree that a further validation may strengthen the paper. However, it is important to consider that the aim of this study is the prediction of chromothripsis in hematological malignancies (see above). This decision is based on our group expertise and on the knowledge that both chromothripsis and the non-chromothripsis genomic background of these cancers is different from solid tumors.

Regarding the importance of chromothripsis prediction in MM, we respectfully disagree with the Reviewer comment regarding appropriateness of our findings for this journal. MM is the second most common hematological cancer, and it is consistently preceded by asymptomatic precursor conditions monoclonal gammopathy of uncertain significance or smoldering MM. These two conditions affect 3% of the population older than 40 years. Only a fraction of these patients will progress to MM, but given the potential catastrophic clinical presentation of MM, definition of precursors at high risk of progression and refinement of early intervention strategies is a critical unmet clinical need.

A few weeks ago we published a paper (Oben et al. Nat Comm 2021; PMID: 33767199) demonstrating how the presence of chromothripsis defines the entity of high-risk precursors, making chromothripsis one of the most important prognostic feature in MM precursor conditions. In addition, chromothripsis is emerging as one of strongest prognostic features in active MM, more relevant than previously defined high-risk FISH-

based features such as t(4;14) and del17p (Rustad et al. Blood Cancer Discovery 2020, PMID: 33392515). In AML, chromothripsis is infrequent and under-investigated. However, in therapy-related AML this complex event is frequently detectable, and its presence usually confer a complex cytogenetic/karyotype which represents one of most unfavorable prognostic features in our clinical practice. In lymphomas, most of the chromothripsis seems to be clustered in the diffuse large B-cell lymphoma group, but their prognostic role has never been tested on large and fully characterized cohort.

Overall, these considerations are aimed to show to the Reviewer that a method for chromothripsis detection in hematological cancers (and particularly MM) is extremely relevant from a translational perspective.

- In figure 3 there are multiple events per sample. In addition to the covariates used in the Cox models, how are these other variables (e.g. ploidy) affecting the results? Were these clinical variables were considered in the 'feature selection' step?

In MM, the baseline ploidy has a bimodal distribution, with one group of patients having a ploidy of 1.8-2.2 and a group (called hyperdiploidy) with ploidy 2.5-3 (Smadja et al. Leukemia 1998, PMID: 9639426). This second group comprises patients harboring multiple whole chromosome (or arm) trisomies involving all the odd chromosomes (aside from 1, 13 and 17). Since the 1990s' we have known that hyperdiploid patients have a distinct gene expression profile and genomic driver landscape (e.g., enrichment of *KRAS* and *NRAS* mutations, no IGH-translocations). These patients (~50% of all MM) tend to have often good outcome. For this reason, instead of the general ploidy, we decided to include in the stepwise regression this key myeloma-specific feature (i.e. hyperdiploidy). Chromothripsis can be present in either hyperdiploidy and non-hyperdiploid cases, and the adverse survival associated with a high CN signature prediction score is retained with the addition of hyperdiploidy to multivariate analysis (see figure below).

Multivariate analysis for progression-free survival

Figure legend: (a) Progression-free survival (PFS) according to targeted-sequencing-based copy number signatures (tCN-SIG2, contribution below the median; red, above the median; blue). (a) Multivariate analysis for PFS according to contribution from tCN-SIG2 after correction for age, International Staging Score (ISS), gain or amplification of 1q21, mutation of TP53 or translocation t(4;14) (t_MMSET).

- In Figure 4a, what are the “6” and “9” in the plot?

In keeping with conventions established in numerous papers from Peter Campbell’s Lab, vertical lines represent SV breakpoints for deletion (red), inversion (blue), tandem-duplication (green) and translocations (black). The number above the black line represents the chromosome involved by the translocation. In this example case we have multiple translocations between chr 16 and each of chr 6 and chr 9. We have added text to clarify in the new Figure 7a legend.

- Regarding the analysis reported in Figure 5, how do these results compare to patient stratification based on other variables related to genome instability, such as the presence of complex rearrangements, percentage of the genome altered, number of SVs, etc? Are the results still significant after including other covariates accounting for genome instability in the model? That is, does the signal from chromothripsis stand out even if ploidy, percentage of genome altered, etc.. are included in the model?

As discussed above, chromothripsis does not appear to reflect presence of genomic instability in MM, given its structural relatively stability over time and the non-complex non-chromothripsis genomic background. Furthermore, all features that in MM are linked to genomic complexity (eg APOBEC activity, TP53 inactivation) are usually acquired later in time. While TP53 and APOBEC are associated with poor prognosis, the presence of complex rearrangements, percentage of the genome altered, or number of SVs are not independently associated with survival (Rustad et al. Blood Cancer Discovery 2020, PMID: 33392515).

To further address the Reviewers comments, we demonstrate that the addition of chromoplexy, complex SVs, the LOH-index, and the genomic scar score into the multivariate analysis of PFS demonstrates each of these not to have a significant association with PFS, while the CN signature-based prediction remains significant (see figure below).

Multivariate analysis for progression-free survival

Figure legend: Multivariate analysis for progression-free survival PFS according to copy number signature prediction score (CN_pred) score after correction for chromoplexy, complex SV, the genomic scar score (GSS), the loss of heterozygosity index (LOH_index), International Staging Score (ISS), age, Eastern Cooperative Oncology Group (ECOG) score, gain or amplification of 1q21 and APOBEC mutational activity.

- Chromothripsis is treated as a binary variable in this study. Although the chromothripsis patterns might be less complex than those observed in other malignancies, e.g. sarcomas, chromothripsis patterns have classified in distinct categories, and pan-cancer analysis of this mechanism have certainly revealed a very complex and varied landscape. Could the authors elaborate more on the types of chromothripsis detected in their cases, further stratify based on timing (e.g. before and after WGD), etc..? Referring to a previous publication is not acceptable given the central role of chromothripsis in this study. Treating chromothripsis as a binary variable is an oversimplification of this mechanism and its consequences.

We agree that references cannot substitute data, but at the same time, in the last two years, we have published at least 6 original articles focused on these distinct features (Oben et al. Nat Comm 2021, PMID: 33767199; Maura et al. Clin Cancer Res 2021, PMID: 33504553; Landau et al. Nat Comm 2021, PMID: 33473129; Rustad et al. Blood Cancer Discovery PMID: 33392515; Maura et al. Nat Comm 2019 PMID: 31444325; Maura et al. Nat Comm 2019, PMID: 31278357), and it would be impossible to re-present all these data. In the new manuscript we have expanded the introduction and the discussion in order to provide a more comprehensive background of MM biology.

Regarding the points raised by the Reviewer:

- Whole genome duplication is extremely rare in MM, and virtually absent at baseline. Cases with whole genome doubling have been reported at relapse, suggesting a late role in MM (Maura et al. Nat Comm 2019; PMID: 31278357).
- We have treated chromothripsis as a binary variable for CN-signature analysis, however in response to Reviewer comments we have added the section looking at false positive / false negative cases, presenting how complexity of the chromothripsis is important for detection. As explained above, both *ShatterSeek* and CN-signatures are able to detect the most complex chromothripsis events. Of relevance these events seem to be the main driver of the poor prognosis associated with chromothripsis.

As extensively discussed above and in the new version of the paper (page 10, lines 213-223), presence of chromothripsis in MM and hematological cancers doesn't equate to genomic instability.

- A major issue of this study is that the description of the methods used is insufficient. For instance: What algorithm was used to call somatic copy number aberrations using the WGS data? How were the copy number and ploidy calls validated? What parameter were chosen to run controlFREEC and how was the 'verification' performed? How was the ploidy set up in controlFREEC? Etc..

All code for quality adjustment and filtering of CN calls is included in Supplementary Data 1. The CN calls were obtained using tCoNuT; a TGen developed tool, with MMRF CoMMpass specific optimizations (<https://github.com/tgen/tCoNuT>; Rustad et al. Blood Cancer Discovery PMID: 33392515). The verification step involved a comparison with calls obtained with controlFREEC (standard settings for GC content / read size / segmentation / windows), which demonstrated >90% agreement between the two approaches. Ultimately, we used the tCoNuT calls as they contained less noise, with the link to standard MM-optimized tCoNuT settings provided in the Methods (page 19, line 427-437).

As described above, MM ploidy has distinct features, with a bimodal distribution comprising near-diploid and hyperdiploid. The extreme CN observed in solid cancers are not evident in MM, with MM ploidy being well captured by standard CN assessment methods.

The methods section related to the selection of the optimal number of categories in each of the 6 CN features using a mixed effect model should be extended to allow future readers to follow/reproduce all steps of the analysis.

The entire code is attached to the paper to allow anyone to reproduce our data (please see Supplementary Data 1 through 4).

The methodology for chromothripsis detection, which is key to his study, is not described in sufficient detail. This should be addressed carefully. The software and parameters used for statistical significance are not included. Details on the validation for the cut-off values used are also missing. Etc. How do their calls compare to other chromothripsis detection methods such as ChromAL (Notta et al 2016) or ShatterSeek (Cortes-Ciriano, Nat Genet 2020). Even if the authors are using data from a previous study this information should be included here.

Similarly, the detection of chromoplexy is defined as: "Chromoplexy was defined by interconnected SV breakpoints across >2 chromosomes associated with CN loss." What algorithm was used? How were these cut-off values validated? While chromoplexy can be associated to copy number loss at the breakpoints forming the chain this is not always the case; how was this handled? How do these results compare to standard chromoplexy detection methods, such as ChainFinder (Baca et al 2013)?

In 2019, together with Peter Campbell (last author), we provided a first overview of SV and complex events in MM (Maura et al. Nat Comm 2019 PMID: 31444325). Subsequently, using the same criteria and approach, we expanded our investigations on 752 long insert low coverage WGS from the CoMMpass data (Rustad et al. Blood cancer Discovery 2020, PMID: 33392515). Overall, as stated in the Method section of the paper, we used previous criteria defined by Peter Campbell, plus we introduced the threshold of 10 or more SVs to define genuine chromothripsis. The complex events with 3-9 SVs defined complex otherwise. These criteria are identical to those recently used by Maciejowski et al. (Nat Gen 2020, PMID: 32719516). As comprehensively described in our previous papers (PMID: 33392515, PMID: 33767199, PMID: 33473129, PMID: 31444325), chromothripsis detection was always confirmed by manual inspection (page 22, lines 484-5), which is regarded as the gold-standard approach by many people involved in the SV field (e.g. Hadi et al. Cell 2020, PMID: 33007263; Cortes-Ciriano et al. Nat Genet 2020, Maciejowski et al. Cell 2015, PMID: 26687355; <https://github.com/parklab/ShatterSeek/blob/master/tutorial.pdf>).

Regarding chromoplexy, algorithms like *ChainFinder* perform quite badly in MM because of the high prevalence of templated insertions (Rustad et al. Blood Cancer Discovery 2020, PMID: 33392515). As a matter of fact, when revising the preliminary chromoplexy prevalence estimate of 18% in the CoMMpass dataset described by the Little Rock UAMS's lab (Blood 2019, <https://doi.org/10.1182/blood-2019-130335>), we observed that more than 80% of these were templated insertions.

Overall, almost 20% of MM patients have a templated insertion involving more than 2 chromosomes. Most of these events involve immunoglobulin loci and create focal gains on oncogene and super enhancer (MYC, CCND1 etc). On the contrary, in MM chromoplexy is mostly linked to chromosomal loss involving onco-suppressors. To differentiate these two events, we use the CN impact of these concatenations and a careful manual inspection with graphical reconstruction.

Overall, we believe that our approach to define complex events in MM is highly accurate and takes into account the unique complexity and structural features of these cancers. In addition, the fact that *ShatterSeek* performed well and similarly to CN-signatures represents a further confirmation of the high quality of our SV calls.

Regarding the comparison between CN-signatures and other methods (i.e. *ShatterSeek*), please see our detailed explanation above.

Minor:

In the Introduction: "Although the biological impact is likely similar across various malignancies,". This claim is not substantiated by any data. A reference is needed or otherwise I suggest removing.

Thank you, we have removed this text.

REVIEWERS' COMMENTS

Reviewer #1 (Remarks to the Author):

The authors have answered comments with satisfaction.

Their CN-signature tool is able to predict the presence of chromothripsis from WES, which has a potential important clinical application in MM - and perhaps in all cancers.

I have no further comments.

Reviewer #2 (Remarks to the Author):

I thank the authors for their detailed response to my comments. My concerns have been successfully addressed, and therefore, I have no additional major comments. However, I think that the authors might want to emphasize further in the main text/abstract that the method has been validated for MM, and that additional validation for the detection of chromothripsis using WES data on other cancer types would require further validation/development.

Re: Copy number signatures predict chromothripsis and clinical outcomes in newly diagnosed multiple myeloma

Thank you for considering our manuscript.

Below, we include a response to the latest reviewer comments – the reviewer comments are in black, our response in blue.

=====

REVIEWERS' COMMENTS

Reviewer #1 (Remarks to the Author):

The authors have answered comments with satisfaction. Their CN-signature tool is able to predict the presence of chromothripsis from WES, which has a potential important clinical application in MM - and perhaps in all cancers. I have no further comments.

Thank you for your kind consideration and comments.

Reviewer #2 (Remarks to the Author):

I thank the authors for their detailed response to my comments. My concerns have been successfully addressed, and therefore, I have no additional major comments. However, I think that the authors might want to emphasize further in the main text/abstract that the method has been validated for MM, and that additional validation for the detection of chromothripsis using WES data on other cancer types would require further validation/development.

Thank you for your comments and suggestions. We have added a sentence regarding additional validation for non-hematological cancers to the end of the discussion (page 18, line 405-7).